# Adversarial Distributional Training for Robust Deep Learning

**Yinpeng Dong**[*], **Zhijie Deng**[*], **Tianyu Pang, Jun Zhu, Hang Su**[†]
Dept. of Comp. Sci. & Tech., Institute for AI, BNRist Center
Tsinghua-Bosch Joint ML Center, THBI Lab, Tsinghua University, Beijing, 100084 China
{dyp17, dzj17, pty17}@mails.tsinghua.edu.cn, {suhangss, dcszj}@mail.tsinghua.edu.cn

## Abstract

Adversarial training (AT) is among the most effective techniques to improve model robustness by augmenting training data with adversarial examples. However, most existing AT methods adopt a specific attack to craft adversarial examples, leading to the unreliable robustness against other unseen attacks. Besides, a single attack algorithm could be insufficient to explore the space of perturbations. In this paper, we introduce adversarial distributional training (ADT), a novel framework for learning robust models. ADT is formulated as a minimax optimization problem, where the inner maximization aims to learn an adversarial distribution to characterize the potential adversarial examples around a natural one under an entropic regularizer, and the outer minimization aims to train robust models by minimizing the expected loss over the worst-case adversarial distributions. Through a theoretical analysis, we develop a general algorithm for solving ADT, and present three approaches for parameterizing the adversarial distributions, ranging from the typical Gaussian distributions to the flexible implicit ones. Empirical results on several benchmarks validate the effectiveness of ADT compared with the state-of-the-art AT methods.

## 1 Introduction

While recent breakthroughs in deep neural networks (DNNs) have led to substantial success in a wide range of fields [21], DNNs also exhibit adversarial vulnerability to small perturbations around the input [60, 22]. Due to the security threat, considerable efforts have been devoted to improving the adversarial robustness of DNNs [22, 36, 38, 42, 68, 46, 72, 47, 78]. Among them, adversarial training (AT) is one of the most effective techniques [2, 16]. AT can be formulated as a minimax optimization problem [42], where the inner maximization aims to find an adversarial example that maximizes the classification loss for a natural one, while the outer minimization aims to train a robust classifier using the generated adversarial examples. To solve the non-concave and typically intractable inner maximization problem approximately, several adversarial attack methods can be adopted, such as fast gradient sign method (FGSM) [22] and projected gradient descent (PGD) method [42].

However, existing AT methods usually solve the inner maximization problem based on a specific attack algorithm, some of which can result in poor generalization for other unseen attacks *under the same threat model* [58][1]. For example, defenses trained on the FGSM adversarial examples, without random initialization or early stopping [69], are vulnerable to multi-step attacks [36, 63]. Afterwards, recent methods [77, 70] can achieve the state-of-the-art robustness against the commonly used attacks (e.g., PGD), but they can still be defeated by others [39, 64]. It indicates that these defenses probably cause gradient masking [63, 2, 66], and can be fooled by stronger or adaptive attacks.

---

[*]Equal contribution. † Corresponding author.
[1]It should be noted that we consider the generalization problem across attacks *under the same threat model*, rather than studying the generalization ability *across different threat models* [24, 17, 62].

Moreover, a single attack algorithm could be insufficient to explore the space of possible perturbations. PGD addresses this issue by using random initialization, however the adversarial examples crafted by PGD with random restarts probably lie together and lose diversity [61]. As one key to the success of AT is how to solve the inner maximization problem, other methods perform training against multiple adversaries [63, 28], which can be seen as more exhaustive approximations of the inner problem [42]. Nevertheless, there still lacks a formal characterization of multiple, diverse adversaries.

To mitigate the aforementioned issues and improve the model robustness against a wide range of adversarial attacks, in this paper we present **adversarial distributional training (ADT)**, a novel framework that explicitly models the adversarial examples around a natural input using a distribution. Subsuming AT as a special case, ADT is formulated as a minimax problem, where the inner maximization aims to find an adversarial distribution for each natural example by maximizing the expected loss over this distribution, while the outer minimization aims to learn a robust classifier by minimizing the expected loss over the worst-case adversarial distributions. To keep the adversarial distribution from collapsing into a Delta one, we explicitly add an entropic regularization term into the objective, making the distribution capable of characterizing heterogeneous adversarial examples.

Through a theoretical analysis, we show that the minimax problem of ADT can be solved sequentially similar to AT [42]. We implement ADT by parameterizing the adversarial distributions with trainable parameters, with three concrete examples ranging from the classical Gaussian distributions to the very flexible implicit density models. Extensive experiments on the CIFAR-10 [34], CIFAR-100 [34], and SVHN [44] datasets validate the effectiveness of our proposed methods on building robust deep learning models, compared with the alternative state-of-the-art AT methods.

## 2 Proposed method

In this section, we first introduce the background of adversarial training (AT), then detail adversarial distributional training (ADT) framework, and finally provide a general algorithm for solving ADT.

### 2.1 Adversarial training

Adversarial training has been widely studied to improve the adversarial robustness of DNNs. Given a dataset $\mathcal{D} = \{(\mathbf{x}_i, y_i)\}_{i=1}^n$ of $n$ training samples with $\mathbf{x}_i \in \mathbb{R}^d$ and $y_i \in \{1, ..., C\}$ being the natural example and the true label, AT can be formulated as a minimax optimization problem [42] as

$$\min_{\boldsymbol{\theta}} \frac{1}{n} \sum_{i=1}^n \max_{\boldsymbol{\delta}_i \in \mathcal{S}} \mathcal{L}(f_{\boldsymbol{\theta}}(\mathbf{x}_i + \boldsymbol{\delta}_i), y_i), \tag{1}$$

where $f_{\boldsymbol{\theta}}$ is the DNN model with parameters $\boldsymbol{\theta}$ that outputs predicted probabilities over all classes, $\mathcal{L}$ is a loss function (e.g., cross-entropy loss), and $\mathcal{S} = \{\boldsymbol{\delta} : \|\boldsymbol{\delta}\|_\infty \leq \epsilon\}$ is a perturbation set with $\epsilon > 0$. This is the $\ell_\infty$ threat model widely studied before and what we consider in this paper. Our method can also be extended to other threat models (e.g., $\ell_2$ norm), which we leave to future work.

This minimax problem is usually solved sequentially, i.e., adversarial examples are crafted by solving the inner maximization first, and then the model parameters are optimized based on the generated adversarial examples. Several attack methods can be used to solve the inner maximization problem approximately, such as FGSM [22] or PGD [42]. For example, PGD takes multiple gradient steps as

$$\boldsymbol{\delta}_i^{t+1} = \Pi_{\mathcal{S}}\big(\boldsymbol{\delta}_i^t + \alpha \cdot \text{sign}(\nabla_{\mathbf{x}} \mathcal{L}(f_{\boldsymbol{\theta}}(\mathbf{x}_i + \boldsymbol{\delta}_i^t), y_i))\big), \tag{2}$$

where $\boldsymbol{\delta}_i^t$ is the adversarial perturbation at the $t$-th step, $\Pi(\cdot)$ is the projection function, and $\alpha$ is a small step size. $\boldsymbol{\delta}_i^0$ is initialized uniformly in $\mathcal{S}$. $\boldsymbol{\delta}_i^t$ will converge to a local maximum eventually.

### 2.2 Adversarial distributional training

As we discussed, though effective, AT is not problemless. AT with a specific attack possibly leads to overfitting on the attack pattern [36, 63, 77, 70], which hinders the trained models from defending against other attacks. And a single attack algorithm may be unable to explore all possible perturbations in the high-dimensional space, which could result in unsatisfactory robustness performance [63, 28].

To alleviate these problems, we propose to capture the distribution of adversarial perturbations around each input instead of only finding a locally most adversarial point for more generalizable adversarial

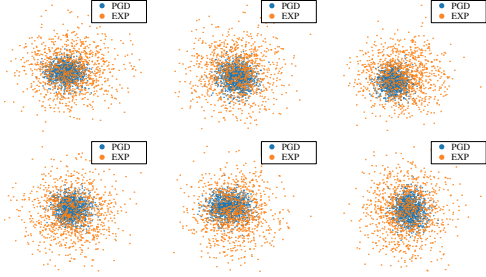

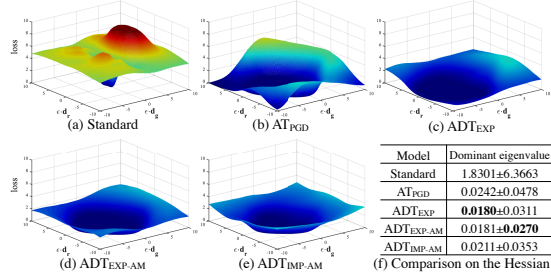

| Model | Dominant eigenvalue |
|---|---|
| Standard | 1.8301±6.3663 |
| $\text{AT}_{\text{PGD}}$ | 0.0242±0.0478 |
| $\text{ADT}_{\text{EXP}}$ | **0.0180**±0.0311 |
| $\text{ADT}_{\text{EXP-AM}}$ | 0.0181±**0.0270** |
| $\text{ADT}_{\text{IMP-AM}}$ | 0.0211±0.0353 |

(f) Comparison on the Hessian

Figure 1: Visualization of the adversarial examples generated by PGD with random restarts (blue) and those sampled from the adversarial distribution learned by $\text{ADT}_{\text{EXP}}$ (orange). Each subfigure corresponds to one randomly selected data point.

Figure 2: Visualization of loss surfaces in the vicinity of an input along the gradient direction ($\mathbf{d_g}$) and a random direction ($\mathbf{d_r}$) for various models in (a)-(e). (f) reports the dominant eigenvalue of the Hessian matrix of the classification loss w.r.t. the input. Full details are in Sec. 5.3.

training, called **adversarial distributional training (ADT)**. In particular, we model the adversarial perturbations around each natural example $\mathbf{x}_i$ by a distribution $p(\boldsymbol{\delta}_i)$, whose support is contained in $\mathcal{S}$. Based on this, ADT is formulated as a distribution-based minimax optimization problem as

$$\min_{\boldsymbol{\theta}} \frac{1}{n} \sum_{i=1}^{n} \max_{p(\boldsymbol{\delta}_i) \in \mathcal{P}} \mathbb{E}_{p(\boldsymbol{\delta}_i)} \big[ \mathcal{L}(f_{\boldsymbol{\theta}}(\mathbf{x}_i + \boldsymbol{\delta}_i), y_i) \big], \tag{3}$$

where $\mathcal{P} = \{p : \text{supp}(p) \subseteq \mathcal{S}\}$ is a set of distributions with support contained in $\mathcal{S}$. As can be seen in Eq. (3), the inner maximization aims to learn an adversarial distribution, such that a point drawn from it is likely an adversarial example. And the objective of the outer minimization is to adversarially train the model parameters by minimizing the expected loss over the worst-case adversarial distributions induced by the inner problem. It is noteworthy that AT is a special case of ADT, by specifying the distribution family $\mathcal{P}$ to contain Delta distributions only.

**Regularizing adversarial distributions.** For the inner maximization of ADT, we can easily see that

$$\max_{p(\boldsymbol{\delta}_i) \in \mathcal{P}} \mathbb{E}_{p(\boldsymbol{\delta}_i)} \big[ \mathcal{L}(f_{\boldsymbol{\theta}}(\mathbf{x}_i + \boldsymbol{\delta}_i), y_i) \big] \leq \max_{\boldsymbol{\delta}_i \in \mathcal{S}} \mathcal{L}(f_{\boldsymbol{\theta}}(\mathbf{x}_i + \boldsymbol{\delta}_i), y_i). \tag{4}$$

It indicates that the optimal distribution by solving the inner problem of ADT will degenerate into a Dirac one. Hence the adversarial distribution cannot cover a diverse set of adversarial examples, and ADT becomes AT. To solve this issue, we add an entropic regularization term into the objective as

$$\min_{\boldsymbol{\theta}} \frac{1}{n} \sum_{i=1}^{n} \max_{p(\boldsymbol{\delta}_i) \in \mathcal{P}} \mathcal{J}\big(p(\boldsymbol{\delta}_i), \boldsymbol{\theta}\big), \text{ with } \mathcal{J}\big(p(\boldsymbol{\delta}_i), \boldsymbol{\theta}\big) = \mathbb{E}_{p(\boldsymbol{\delta}_i)} \big[ \mathcal{L}(f_{\boldsymbol{\theta}}(\mathbf{x}_i + \boldsymbol{\delta}_i), y_i) \big] + \lambda \mathcal{H}(p(\boldsymbol{\delta}_i)), \tag{5}$$

where $\mathcal{H}(p(\boldsymbol{\delta}_i)) = -\mathbb{E}_{p(\boldsymbol{\delta}_i)}[\log p(\boldsymbol{\delta}_i)]$ is the entropy of $p(\boldsymbol{\delta}_i)$, $\lambda$ is a balancing hyperparameter, and $\mathcal{J}\big(p(\boldsymbol{\delta}_i), \boldsymbol{\theta}\big)$ denotes the overall loss function for notation simplicity. Note that the entropy maximization is a common technique to increase the support of a distribution in generative modeling [12, 13] or reinforcement learning [23]. We next discuss why ADT is superior to AT.

### 2.2.1 Discussion on the superiority of ADT

The major difference between AT and ADT is that for each natural input $\mathbf{x}_i$, AT finds a worst-case adversarial example, while ADT learns a worst-case adversarial distribution comprising a variety of adversarial examples. Because adversarial examples can be generated by various attacks, we expect that those adversarial examples probably lie in the region where the adversarial distribution assigns high probabilities, such that minimizing the expected loss over this distribution can naturally lead to a better generalization ability of the trained classifier across attacks under the same threat model.

Furthermore, as we add an entropic regularizer into the objective (5), the adversarial distribution is able to better explore the space of possible perturbations and characterize more diverse adversarial examples compared with a single attack method (e.g., PGD). To show this, for each data we generate a set of adversarial examples by PGD with random restarts and sample another set of adversarial examples from the adversarial distribution learned by $\text{ADT}_{\text{EXP}}$ (a variant of ADT detailed in Sec. 3.1),

**Algorithm 1** The general algorithm for ADT

---
**Input:** Training data $\mathcal{D}$, objective function $\mathcal{J}\big(p(\boldsymbol{\delta}_i), \boldsymbol{\theta}\big)$, the set of perturbation distributions $\mathcal{P}$, training epochs $N$, and learning rate $\eta$.
1: Initialize $\boldsymbol{\theta}$;
2: **for** epoch = 1 **to** $N$ **do**
3:     **for** each minibatch $\mathcal{B} \subset \mathcal{D}$ **do**
4:         Obtain $p^*(\boldsymbol{\delta}_i)$ for each input $(\mathbf{x}_i, y_i) \in \mathcal{B}$ by solving $p^*(\boldsymbol{\delta}_i) = \arg\max_{p(\boldsymbol{\delta}_i) \in \mathcal{P}} \mathcal{J}\big(p(\boldsymbol{\delta}_i), \boldsymbol{\theta}\big)$;
5:         Update $\boldsymbol{\theta}$ with stochastic gradient descent $\boldsymbol{\theta} \leftarrow \boldsymbol{\theta} - \eta \cdot \mathbb{E}_{(\mathbf{x}_i, y_i) \in \mathcal{B}}\big[\nabla_{\boldsymbol{\theta}} \mathcal{J}\big(p^*(\boldsymbol{\delta}_i), \boldsymbol{\theta}\big)\big]$;
6:     **end for**
7: **end for**

---

targeted at a standard trained model. Then we can visualize these adversarial examples by projecting them onto the 2D space spanned by the first two eigenvectors given by PCA [30]. The visualization results of some randomly selected data points in Fig. 1 show that adversarial examples sampled from the adversarial distribution are scattered while those crafted by PGD concentrate together. We further evaluate the diversity of adversarial examples by quantitatively measuring their average pairwise distances. The average $\ell_2$ distance of adversarial examples sampled from the adversarial distribution over $100$ test images is $1.95$, which is $1.56$ for PGD. Although the adversarial distributions can characterize more diverse adversarial examples, they have a similar attack power compared with PGD, as later shown in Table 4. Minimizing the loss on such diverse adversarial examples can consequently help to learn a smoother and more flattened loss surface around the natural examples in the input space, as shown in Fig. 2. Therefore, ADT can improve the overall robustness compared with AT.

## 2.3 A general algorithm for ADT

To solve minimax problems, Danskin's theorem [14] states how the maximizers of the inner problem can be used to define the gradients for the outer problem, which is also the theoretical foundation of AT [42]. However, it is problematic to directly apply Danskin's theorem for solving ADT since the search space $\mathcal{P}$ may not be compact, which is one assumption of this theorem. As it is non-trivial to perform a theoretical analysis on how to solve ADT, we first lay out the following assumptions.

**Assumption 1.** *The loss function $\mathcal{J}\big(p(\boldsymbol{\delta}_i), \boldsymbol{\theta}\big)$ is continuously differentiable w.r.t. $\boldsymbol{\theta}$.*

Assumption 1 is also made in [42] for AT. Although the loss function is not completely continuously differentiable due to the ReLU layers, the set of discontinuous points has measure zero, such that it is assumed not to be an issue in practice [42].

**Assumption 2.** *Probability density functions of distributions in $\mathcal{P}$ are bounded and equicontinuous.*

Assumption 2 puts a restriction on the set of distributions $\mathcal{P}$. We show that the explicit adversarial distributions proposed in Sec. 3.1 satisfy this assumption (in Appendix B.1).

**Theorem 1.** *Suppose Assumptions 1 and 2 hold. We define $\rho(\boldsymbol{\theta}) = \max_{p(\boldsymbol{\delta}_i) \in \mathcal{P}} \mathcal{J}\big(p(\boldsymbol{\delta}_i), \boldsymbol{\theta}\big)$, and $\mathcal{P}^*(\boldsymbol{\theta}) = \{p(\boldsymbol{\delta}_i) \in \mathcal{P} : \mathcal{J}\big(p(\boldsymbol{\delta}_i), \boldsymbol{\theta}\big) = \rho(\boldsymbol{\theta})\}$. Then $\rho(\boldsymbol{\theta})$ is directionally differentiable, and its directional derivative along the direction $\mathbf{v}$ satisfies*

$$\rho'(\boldsymbol{\theta}; \mathbf{v}) = \sup_{p(\boldsymbol{\delta}_i) \in \mathcal{P}^*(\boldsymbol{\theta})} \mathbf{v}^\top \nabla_{\boldsymbol{\theta}} \mathcal{J}\big(p(\boldsymbol{\delta}_i), \boldsymbol{\theta}\big). \tag{6}$$

*Particularly, when $\mathcal{P}^*(\boldsymbol{\theta}) = \{p^*(\boldsymbol{\delta}_i)\}$ only contains one maximizer, $\rho(\boldsymbol{\theta})$ is differentiable at $\boldsymbol{\theta}$ and*

$$\nabla_{\boldsymbol{\theta}} \rho(\boldsymbol{\theta}) = \nabla_{\boldsymbol{\theta}} \mathcal{J}\big(p^*(\boldsymbol{\delta}_i), \boldsymbol{\theta}\big). \tag{7}$$

The complete proof of Theorem 1 is deferred to Appendix B.1. Theorem 1 provides us a general principle for training ADT, by first solving the inner problem and then updating the model parameters along the gradient direction of the loss function at the global maximizer of the inner problem, in a sequential manner similar to AT [42]. We provide the general algorithm for ADT in Alg. 1. Analogous to AT, the global maximizer of the inner problem cannot be solved analytically. Therefore, we propose three different approaches to obtain approximate solutions, as introduced in Sec. 3. Although we cannot reach the global maximizer of the inner problem, our experiments suggest that we can reliably solve the minimax problem (5) by our algorithm.

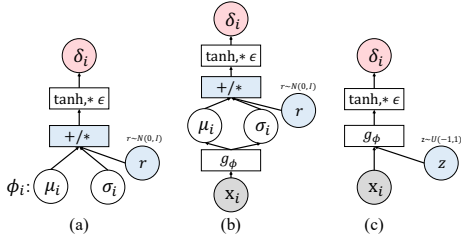

Figure 3: An illustration of the three different approaches to parameterize the distributions of adversarial perturbations. (a) ADT$_{\text{EXP}}$: the explicit adversarial distribution $p_{\phi_i}(\boldsymbol{\delta}_i)$ is defined by transforming $\mathcal{N}(\boldsymbol{\mu}_i, \operatorname{diag}(\boldsymbol{\sigma}_i^2))$ via $\tanh$ followed by a multiplication with $\epsilon$. (b) ADT$_{\text{EXP-AM}}$: we amortize the explicit adversarial distributions by a neural network $g_{\boldsymbol{\phi}}$ taking $\mathbf{x}_i$ as input. (c) ADT$_{\text{IMP-AM}}$: we define the implicit adversarial distributions by inputting an additional random variable $\mathbf{z} \sim \mathrm{U}(-1, 1)$ to the network $g_{\boldsymbol{\phi}}$.

# 3 Parameterizing adversarial distributions

At the core of ADT lie the solutions of the inner maximization problem of Eq. (5). The basic idea is to parameterize the adversarial distributions with trainable parameters $\boldsymbol{\phi}_i$. With the parameterized $p_{\boldsymbol{\phi}_i}(\boldsymbol{\delta}_i)$, the inner problem is converted into maximizing the expected loss w.r.t. $\boldsymbol{\phi}_i$. In the following, we present the parametrizations and learning strategies of three different approaches, respectively. We provide an overview of these approaches in Fig. 3.

## 3.1 ADT$_{\text{EXP}}$: explicit modeling of adversarial perturbations

A natural way to model adversarial perturbations around an input data is using a distribution with an explicit density function. We name ADT with EXPlicit adversarial distributions as ADT$_{\text{EXP}}$. To define a proper distribution $p_{\boldsymbol{\phi}_i}(\boldsymbol{\delta}_i)$ on $\mathcal{S}$, we take the transformation of random variable approach as

$$\boldsymbol{\delta}_i = \epsilon \cdot \tanh(\mathbf{u}_i), \quad \mathbf{u}_i \sim \mathcal{N}(\boldsymbol{\mu}_i, \operatorname{diag}(\boldsymbol{\sigma}_i^2)), \tag{8}$$

where $\mathbf{u}_i$ is sampled from a diagonal Gaussian distribution with $\boldsymbol{\mu}_i, \boldsymbol{\sigma}_i \in \mathbb{R}^d$ as the mean and standard deviation. $\mathbf{u}_i$ is transformed by a $\tanh$ function and then multiplied by $\epsilon$ to get $\boldsymbol{\delta}_i$. We let $\boldsymbol{\phi}_i = (\boldsymbol{\mu}_i, \boldsymbol{\sigma}_i)$ denote the parameters to be learned. We sample $\mathbf{u}_i$ from a diagonal Gaussian mainly for the sake of computational simplicity. But our method is fully compatible with more expressive distributions, such as matrix-variate Gaussians [40] or multiplicative normalizing flows [41], and we leave using them to future work. Given Eq. (8), the inner problem of Eq. (5) becomes

$$\max_{\boldsymbol{\phi}_i} \left\{ \mathbb{E}_{p_{\boldsymbol{\phi}_i}(\boldsymbol{\delta}_i)} \big[ \mathcal{L}(f_{\boldsymbol{\theta}}(\mathbf{x}_i + \boldsymbol{\delta}_i), y_i) \big] + \lambda \mathcal{H}(p_{\boldsymbol{\phi}_i}(\boldsymbol{\delta}_i)) \right\}. \tag{9}$$

To solve this, we need to estimate the gradient of the expected loss w.r.t. the parameters $\boldsymbol{\phi}_i$. A commonly used method is the low-variance reparameterization trick [33, 5], which replaces the sampling process of the random variable of interest with the corresponding differentiable transformation. With this technique, the gradient can be back-propagated from the samples to the distribution parameters directly. In our case, we reparameterize $\boldsymbol{\delta}_i$ by $\boldsymbol{\delta}_i = \epsilon \cdot \tanh(\mathbf{u}_i) = \epsilon \cdot \tanh(\boldsymbol{\mu}_i + \boldsymbol{\sigma}_i \mathbf{r})$, where $\mathbf{r}$ is an auxiliary noise variable following the standard Gaussian distribution $\mathcal{N}(\mathbf{0}, \mathbf{I})$. Therefore, we can estimate the gradient of $\boldsymbol{\phi}_i$ via

$$\mathbb{E}_{\mathbf{r} \sim \mathcal{N}(\mathbf{0}, \mathbf{I})} \nabla_{\boldsymbol{\phi}_i} \Big[ \mathcal{L}\big(f_{\boldsymbol{\theta}}\big(\mathbf{x}_i + \epsilon \cdot \tanh(\boldsymbol{\mu}_i + \boldsymbol{\sigma}_i \mathbf{r})\big), y_i\big) - \lambda \log p_{\boldsymbol{\phi}_i}\big(\epsilon \cdot \tanh(\boldsymbol{\mu}_i + \boldsymbol{\sigma}_i \mathbf{r})\big) \Big]. \tag{10}$$

The first term inside is the classification loss with the sampled noise, and the second is the negative log density (i.e., estimation of entropy). It can be calculated analytically (proof in Appendix B.2) as

$$\sum_{j=1}^{d} \Big( \frac{1}{2}(\mathbf{r}^{(j)})^2 + \frac{\log 2\pi}{2} + \log \boldsymbol{\sigma}_i^{(j)} + \log\big(1 - \tanh(\boldsymbol{\mu}_i^{(j)} + \boldsymbol{\sigma}_i^{(j)} \mathbf{r}^{(j)})^2\big) + \log \epsilon \Big), \tag{11}$$

where the superscript $j$ denotes the $j$-th element of a vector.

In practice, we approximate the expectation in Eq. (10) with $k$ Monte Carlo (MC) samples, and perform $T$ steps of gradient ascent on $\boldsymbol{\phi}_i$ to solve the inner problem. After obtaining the optimal parameters $\boldsymbol{\phi}_i^*$, we use the adversarial distribution $p_{\boldsymbol{\phi}_i^*}(\boldsymbol{\delta}_i)$ to update model parameters $\boldsymbol{\theta}$.

## 3.2 ADT$_{\text{EXP-AM}}$: amortizing the explicit adversarial distributions

Although the aforementioned method in Sec. 3.1 provides a simple way to learn explicit adversarial distributions for ADT, it needs to learn the distribution parameters for each input and then brings

prohibitive computational cost. Compared with PGD-based AT which constructs adversarial examples by $T$ steps PGD [42], $\text{ADT}_{\text{EXP}}$ is approximately $k$ times slower since the gradient of $\phi_i$ is estimated by $k$ MC samples in each step. In this subsection, we propose to amortize the inner optimization of $\text{ADT}_{\text{EXP}}$, to develop a more feasible and scalable training method. We name ADT with the AMortized version of EXPlicit adversarial distributions as $\text{ADT}_{\text{EXP-AM}}$.

Instead of learning the distribution parameters for each data $\mathbf{x}_i$, we opt to learn a mapping $g_{\boldsymbol{\phi}} : \mathbb{R}^d \rightarrow \mathcal{P}$, which defines the adversarial distribution for each input in a conditional manner $p_{\boldsymbol{\phi}}(\boldsymbol{\delta}_i|\mathbf{x}_i)$. We instantiate $g_{\boldsymbol{\phi}}$ by a conditional generator network. It takes a natural example $\mathbf{x}_i$ as input, and outputs the parameters $(\boldsymbol{\mu}_i, \boldsymbol{\sigma}_i)$ of its corresponding explicit adversarial distribution, which is also defined by Eq. (8). The advantage of this method is that the generator network can potentially learn common structures of the adversarial perturbations, which can generalize to other training samples [3, 49]. It means that we do not need to optimize $\phi$ excessively on each data $\mathbf{x}_i$, which can accelerate training.

### 3.3  $\text{ADT}_{\text{IMP-AM}}$: implicit modeling of adversarial perturbations

Since the underlying distributions of adversarial perturbations have not been figured out yet and could be different across samples, it is hard to specify a proper explicit distribution of adversarial examples, which may lead to the underfitting problem. To bypass this, we resort to implicit distributions (i.e., distributions without tractable probability density functions but can still be sampled from), which have shown promising results recently [20, 55, 56], particularly in modeling complex high-dimensional data [51, 27]. The major advantage of implicit distributions is that they are not confined to provide explicit densities, which improves the flexibility inside the sampling process.

Based on this, we propose to use the implicit distributions to characterize the adversarial perturbations. Considering the priority of amortized optimization, we learn a generator $g_{\boldsymbol{\phi}} : \mathbb{R}^{d_z} \times \mathbb{R}^d \rightarrow \mathbb{R}^d$ which implicitly defines a conditional distribution $p_{\boldsymbol{\phi}}(\boldsymbol{\delta}_i|\mathbf{x}_i)$ by $\boldsymbol{\delta}_i = g_{\boldsymbol{\phi}}(\mathbf{z}; \mathbf{x}_i)$, where $\mathbf{x}_i$ is a natural input and $\mathbf{z} \in \mathbb{R}^{d_z}$ is a random noise vector. Typically, $\mathbf{z}$ is sampled from a prior $p(\mathbf{z})$ such as the standard Gaussian or uniform distributions as in the generative adversarial networks (GANs) [20]. In this work, we sample $\mathbf{z}$ from a uniform distribution $\text{U}(-1, 1)$. We refer to this approach as $\text{ADT}_{\text{IMP-AM}}$. A practical problem remains unaddressed is that the entropy of the implicit distributions cannot be estimated exactly as we have no access to the density $p_{\boldsymbol{\phi}}(\boldsymbol{\delta}_i|\mathbf{x}_i)$. We instead maximize the variational lower bound of the entropy [12] for its simplicity and success in GANs [13]. We provide full technical details of $\text{ADT}_{\text{EXP-AM}}$ and $\text{ADT}_{\text{IMP-AM}}$, and training algorithms of the three methods in Appendix A.

## 4  Related work

Adversarial machine learning is an emerging research topic with various attack and defense methods being proposed [22, 35, 6, 15, 38, 42, 68, 10, 47]. Besides PGD-based AT [42], recent improvements upon it include designing new losses [78, 43, 48, 50] or network architecture [72], accelerating the training procedure [54, 76, 69], and exploiting more data [25, 1, 8, 75].

Learning the distributions of adversarial examples has been studied before, mainly for black-box adversarial attacks. An adversarial example can be searched over a distribution [26, 37], similar to the inner problem of Eq. (3). But their gradient estimator based on natural evolution strategy exhibits very high variance [33] compared with ours in Eq. (10), since our methods are based on the white-box setting (i.e., compute the gradient) rather than the black-box setting. To the best of our knowledge, we are the first to train robust models by learning the adversarial distributions.

In this work, we adopt a generator network to amortize the adversarial distributions for accelerating the training process. There also exists previous work on adopting generator-based approaches for adversarial attacks and defenses [3, 49, 71]. The inner maximization problem of AT can be solved by generating adversarial examples using a generator network [67, 9], which is similar to our work. The essential difference is that they still focus on the minimax formulation (1) of AT, while we propose a novel ADT framework in Eq. (3). We empirically compare our method with [9] in Appendix D.5.

Adversarial robustness is also related to robustness to certain types of input-agnostic distributions [19]. A classifier robust to Gaussian noise can be turned into a new smoothed classifier that is certifiably robust to $\ell_2$ adversarial examples [11]. Salman et al. [53] further employ adversarial training to improve the certified robustness of randomized smoothing, whereas our method belongs to empirical defenses, aiming to train a robust classifier with the input-dependent adversarial distributions.

Table 1: Classification accuracy of the three proposed methods and baselines on CIFAR-10 under white-box attacks with $\epsilon = 8/255$. The last column shows the overall robustness of the models. We mark the best results for each attack and the overall results that outperform the baselines in **bold**, and the overall best result in <span style="color:blue">blue</span>. We highlight the results of AT$_{FGSM}$ and FeaScatter in <span style="color:orange">orange</span> to emphasize that these models have the generalization problem across attacks, whose overall robustness is weak.

| Model | $\mathcal{A}_{nat}$ | FGSM | PGD-20 | PGD-100 | MIM | C&W | FeaAttack | $\mathcal{A}_{rob}$ |
|---|---|---|---|---|---|---|---|---|
| Standard | **94.81%** | 12.05% | 0.00% | 0.00% | 0.00% | 0.00% | 0.00% | 0.00% |
| AT$_{FGSM}$ | 93.80% | 79.86% | 0.12% | 0.04% | 0.06% | 0.13% | 0.01% | 0.01% |
| AT$_{PGD}$$^\dagger$ | 87.25% | 56.04% | 45.88% | 45.33% | 47.15% | 46.67% | 46.01% | 44.89% |
| AT$_{PGD}$ | 86.91% | 58.30% | 50.03% | 49.40% | 51.40% | 50.23% | 50.46% | 48.26% |
| ALP | 86.81% | 56.83% | 48.97% | 48.60% | 50.13% | 49.10% | 48.51% | 47.90% |
| FeaScatter | 89.98% | 77.40% | 70.85% | 68.81% | 72.74% | 58.46% | 37.45% | 37.40% |
| ADT$_{EXP}$ | 86.89% | 60.41% | 52.18% | **51.69%** | **53.27%** | 52.49% | **52.38%** | **50.56%** |
| ADT$_{EXP-AM}$ | 87.82% | 62.42% | 51.95% | 51.26% | 52.99% | 51.75% | 52.04% | **50.04%** |
| ADT$_{IMP-AM}$ | 88.00% | **64.89%** | **52.28%** | 51.23% | 52.64% | **52.65%** | 51.89% | 49.81% |

The proposed ADT framework is essentially different from a seemingly similar concept — distributionally robust optimization (DRO) [4, 18, 57]. DRO seeks a model that is robust against changes in data-generating distribution, by training on the worst-case data distribution under a probability measure. DRO is related to AT with the Wasserstein distance [57, 59]. However, ADT does not model the changes in data distribution but aims to find an adversarial distribution for each input.

## 5 Experiments

**Experimental settings and implementation details.**[2] We briefly introduce the experimental settings here, and leave the full details in Appendix C. **(A) Datasets:** We perform experiments on the CIFAR-10 [34], CIFAR-100 [34], and SVHN [44] datasets. The input images are normalized to $[0, 1]$. We set the perturbation budget $\epsilon = 8/255$ on CIFAR, and $\epsilon = 4/255$ on SVHN as in [8]. **(B) Network Architectures:** We use a Wide ResNet (WRN-28-10) model [74] as the classifier in most of our experiments following [42]. For the generator network used in ADT$_{EXP-AM}$ and ADT$_{IMP-AM}$, we adopt a popular image-to-image architecture with residual blocks [29, 79]. **(C) Training Details:** We adopt the cross-entropy loss as $\mathcal{L}$ in our objective (5). We set $\lambda = 0.01$ for the entropy term, and leave the study of the effects of $\lambda$ in Sec. 5.3. For ADT$_{EXP}$, we adopt Adam [32] for optimizing $\phi_i$ with the learning rate 0.3, the optimization steps $T = 7$, and the number of MC samples in each step $k = 5$. For ADT$_{EXP-AM}$ and ADT$_{IMP-AM}$, we use $k = 1$ for each data for gradient estimation. **(D) Baselines:** We adopt two primary baselines: 1) standard training on the natural images (***Standard***); 2) AT on the PGD adversarial examples (***AT$_{PGD}$***) [42]. On CIFAR-10, we further incorporate: 1) the pretrained AT$_{PGD}$ model (***AT$_{PGD}$***$^\dagger$) released by [42]; 2) AT on the targeted FGSM adversarial examples (***AT$_{FGSM}$***) [36]; 3) adversarial logit pairing (***ALP***) [31]; and 4) feature scattering-based AT (***FeaScatter***) [77]. We further compare with ***TRADES*** [78] in Sec. 5.4. **(E) Robustness Evaluation:** To evaluate the adversarial robustness of these models, we adopt a plenty of attack methods **A**, and report the *per-example accuracy* as suggested in [7], which calculates the robust accuracy by

$$\mathcal{A}_{rob} = \frac{1}{n_{test}} \sum_{i=1}^{n_{test}} \min_{a \in \mathbf{A}} \mathbb{I}\big( \arg\max\{f_{\boldsymbol{\theta}}(a(\mathbf{x}_i))\} = y_i \big), \tag{12}$$

where $a(\mathbf{x}_i)$ is the adversarial example given by attack $a$, and $\mathbb{I}(\cdot)$ is the indicator function.

### 5.1 Robustness under white-box attacks

We first compare the robustness of the proposed methods with baselines under various white-box attacks. We adopt FGSM [22], PGD [42], MIM [15], C&W [6], and a feature attack (FeaAttack) [39] for evaluation. C&W is implemented by adopting the margin-based loss function in [6] and using PGD for optimization. We use 20 and 100 steps for PGD, 20 steps for MIM, and 30 steps for C&W. The step size is $\alpha = \epsilon/4$ in these attacks. The details of FeaAttack are provided in Appendix C.5.

On **CIFAR-10**, we show the classification accuracy of the proposed methods — ADT$_{EXP}$, ADT$_{EXP-AM}$, ADT$_{IMP-AM}$, and baseline models — Standard, AT$_{FGSM}$, AT$_{PGD}$$^\dagger$, AT$_{PGD}$, ALP, FeaScatter on natural inputs and adversarial examples in Table 1. It is obvious that some AT-based methods exhibit the

Table 2: Classification accuracy of the three proposed methods and baselines on CIFAR-100 and SVHN under part of white-box attacks. Full accuracy results on all adopted white-box attacks can be found in Appendix D.1.

| Model | $\mathcal{A}_{\text{nat}}$ | PGD-20 | PGD-100 | $\mathcal{A}_{\text{rob}}$ | | Model | $\mathcal{A}_{\text{nat}}$ | PGD-20 | PGD-100 | $\mathcal{A}_{\text{rob}}$ |
|---|---|---|---|---|---|---|---|---|---|---|
| Standard | **78.59%** | 0.02% | 0.01% | 0.00% | | Standard | **96.12%** | 3.64% | 2.95% | 2.14% |
| $AT_{\text{PGD}}$ | 61.45% | 25.71% | 25.40% | 24.49% | | $AT_{\text{PGD}}$ | 95.07% | 74.22% | 73.79% | 73.38% |
| $ADT_{\text{EXP}}$ | 62.70% | 28.96% | **28.60%** | 27.13% | | $ADT_{\text{EXP}}$ | 95.70% | **77.01%** | **76.62%** | 75.55% |
| $ADT_{\text{EXP-AM}}$ | 62.84% | 29.01% | 28.46% | **26.87%** | | $ADT_{\text{EXP-AM}}$ | 95.67% | 76.12% | 75.58% | **75.00%** |
| $ADT_{\text{IMP-AM}}$ | 64.07% | **29.40%** | 28.43% | **26.80%** | | $ADT_{\text{IMP-AM}}$ | 95.62% | 75.61% | 74.85% | **74.13%** |

(a) CIFAR-100, $\epsilon = 8/255$.        (b) SVHN, $\epsilon = 4/255$.

Table 3: Accuracy on CIFAR-10 under SPSA attack with different batch sizes and $\epsilon = 8/255$.

| Model | $SPSA_{256}$ | $SPSA_{512}$ | $SPSA_{1024}$ | $SPSA_{2048}$ |
|---|---|---|---|---|
| Standard | 0.00% | 0.00% | 0.00% | 0.00% |
| $AT_{\text{PGD}}$ | 60.67% | 58.10% | 55.82% | 54.37% |
| $ADT_{\text{EXP}}$ | 62.22% | 59.94% | **57.97%** | **56.27%** |
| $ADT_{\text{EXP-AM}}$ | **62.58%** | **60.12%** | 57.62% | 55.84% |
| $ADT_{\text{IMP-AM}}$ | 62.49% | 59.77% | 57.34% | 55.67% |

Table 4: Accuracy on CIFAR-10 under PGD-20, EXP, EXP-AM, and IMP-AM attacks with $\epsilon = 8/255$.

| Model | PGD-20 | EXP | EXP-AM | IMP-AM |
|---|---|---|---|---|
| Standard | 0.00% | 0.00% | 9.24% | 9.83% |
| $AT_{\text{PGD}}$ | 50.03% | 49.97% | 50.46% | 50.36% |
| $ADT_{\text{EXP}}$ | 52.18% | 51.96% | 52.71% | 52.82% |
| $ADT_{\text{EXP-AM}}$ | 51.95% | 51.62% | 52.85% | 52.72% |
| $ADT_{\text{IMP-AM}}$ | 52.28% | 51.46% | 52.76% | 52.48% |

generalization problem across attacks, i.e., $AT_{\text{FGSM}}$ and FeaScatter, whose overall robustness is weak. But ADT-based methods do not have this issue by showing consistent robustness performance across all tested attacks. Although $AT_{\text{PGD}}$ does not have this issue also, and achieves the best performance among the AT-based defenses, the proposed ADT reveals improved overall robustness than $AT_{\text{PGD}}$, showing the effectiveness. We show the results on **CIFAR-100** and **SVHN** in Table 2. The results consistently demonstrate that ADT-based methods can outperform $AT_{\text{PGD}}$ under white-box attacks.

It can be further seen that $ADT_{\text{EXP}}$ is better than $ADT_{\text{EXP-AM}}$ and $ADT_{\text{IMP-AM}}$ in most cases. We suspect the reason is that amortizing the adversarial distributions through a generator network is hard to learn appropriate adversarial regions for every input, owing to the limited capacity of the generator. Nevertheless, it can accelerate training, as shown in Appendix D.4. Note that $ADT_{\text{IMP-AM}}$ obtains similar robustness with $ADT_{\text{EXP-AM}}$. It indicates that though the adopted implicit distributions enable us to optimize in a larger distribution family and the optimization always converges to local optima, $ADT_{\text{IMP-AM}}$ does not necessarily lead to better adversarial distributions and more robust models.

## 5.2 Robustness under black-box attacks

Now we evaluate the robustness of the defenses on CIFAR-10 under black-box attacks to perform a thorough evaluation [7]. We first evaluate *transfer-based black-box attacks* using PGD-20 and MIM. The results in Fig. 4 show that these models obtain higher accuracy under transfer-based attacks than white-box attacks. We further perform *query-based black-box attacks* using SPSA [66] and report the results in Table 3. To estimate the gradients, we set the batch size as 256, 512, 1024, and 2048, the perturbation size as 0.001, and the learning rate as 0.01. We run SPSA attacks for 100 iterations, and early-stop

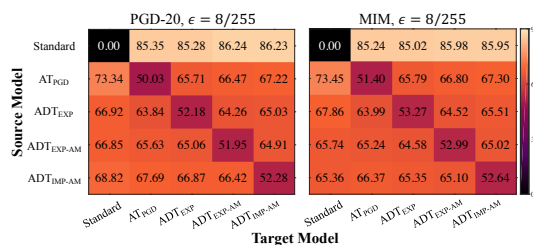

Figure 4: Classification accuracy (%) under transfer-based black-box attacks. The *source model* refers to the one used to craft adversarial examples, and the *target model* is the one being attacked.

when we cause misclassification. The accuracy under SPSA is higher than that under white-box attacks. And our methods obtain better robustness over $AT_{\text{PGD}}$. In summary, the black-box results verify that our methods can reliably improve the robustness rather than causing gradient masking [2].

## 5.3 Additional results and ablation studies

**Attack performance of adversarial distributions.** First, we explore the attack performance of the three proposed methods (i.e., EXP, EXP-AM, and IMP-AM) for learning the adversarial distributions. For EXP, we set $T = 20$, $k = 10$ to conduct a more powerful attack. We further study the convergence of EXP in Appendix D.3. For EXP-AM and IMP-AM, we retrain the generator networks for each pretrained defense. The attack results on five models are shown in Table 4. From the results, EXP is slightly stronger than PGD-20 while EXP-AM and IMP-AM exhibit comparable attack power.

Table 5: Comparison of ADT with TRADES on CIFAR-10. $\beta$ is a hyperparameter balancing the trade-off between natural and robust accuracy. The results of TRADES are reproduced based on the official open-sourced code of [78].

| Model | $\beta$ | $\mathcal{A}_{\mathrm{nat}}$ | PGD-20 | PGD-100 | $\mathcal{A}_{\mathrm{rob}}$ | Model | $\beta$ | $\mathcal{A}_{\mathrm{nat}}$ | PGD-20 | PGD-100 | $\mathcal{A}_{\mathrm{rob}}$ |
|---|---|---|---|---|---|---|---|---|---|---|---|
| TRADES | 1.0 | 87.99% | 51.08% | 48.41% | 47.75% | TRADES | 6.0 | 84.02% | 56.06% | 54.49% | 52.64% |
| ADT$_{\mathrm{EXP}}$ | 1.0 | **89.74%** | 52.39% | 49.88% | **49.05%** | ADT$_{\mathrm{EXP}}$ | 6.0 | 84.66% | 57.71% | **56.17%** | 54.21% |
| ADT$_{\mathrm{EXP-AM}}$ | 1.0 | 88.86% | **54.44%** | **51.66%** | 50.78% | ADT$_{\mathrm{EXP-AM}}$ | 6.0 | 84.85% | 57.67% | 55.73% | **54.09%** |
| ADT$_{\mathrm{IMP-AM}}$ | 1.0 | 88.80% | 54.22% | 51.09% | **50.14%** | ADT$_{\mathrm{IMP-AM}}$ | 6.0 | **84.96%** | **57.82%** | 55.45% | **53.66%** |

**The impact of $\lambda$.** We study the impact of $\lambda$ on the performance of ADT. We choose ADT$_{\mathrm{EXP-AM}}$ as a case study for its fast training process and analytical entropy estimation. Fig. 5 shows the robustness under white-box attacks and the average entropy of the adversarial distributions of ADT$_{\mathrm{EXP-AM}}$ trained with $\lambda = 0.0, 0.001, 0.01, 0.1$, and $1.0$. Generally, a larger $\lambda$ leads to a larger entropy and better robustness. But a too large $\lambda$ will reduce the robustness.

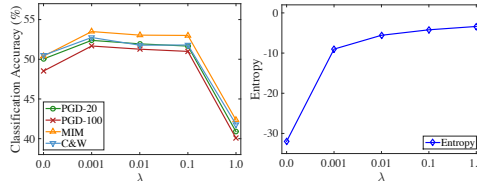

Figure 5: Classification accuracy (%) under white-box attacks and the average entropy of the adversarial distributions of ADT$_{\mathrm{EXP-AM}}$ on CIFAR-10 trained with $\lambda = 0.0, 0.001, 0.01, 0.1$, and $1.0$.

**Loss landscape analysis.** First, we plot the cross-entropy loss of the models projected along the gradient direction ($\mathbf{d_g}$) and a random direction ($\mathbf{d_r}$) in the vicinity of a natural input in Fig. 2. Notably, the models trained by ADT exhibit smoother and more flattened loss surfaces than Standard and AT$_{\mathrm{PGD}}$, and thus deliver better robustness. We further quantitatively measure the smoothness of loss surfaces with the dominant eigenvalue of the Hessian matrix of the classification loss w.r.t. the input as a proxy. We use $1000$ images from the test set of CIFAR-10 for calculation, and report the mean and standard derivation in Fig. 2(f). The numbers are consistent with the visualization results and help us confirm the superiority of ADT upon AT to learn smooth loss surfaces and robust deep models.

## 5.4 Compare with the state-of-the-art

Although we use the cross-entropy loss as our training objective in previous experiments, our proposed ADT framework is compatible with other loss functions. In this section, we integrate TRADES [78], a state-of-the-art AT method, with ADT. We implement ADT by using the TRADES loss in Eq. (5). We follow the same experimental settings as in [78], where a WRN-34-10 model is used and $\epsilon$ is $0.031$. We evaluate the robustness by all adopted attacks. We show part of the results in Table 5, and leave full results in Appendix D.2, which prove that the proposed methods also outperform TRADES.

Besides, a recent state-of-the-art AT$_{\mathrm{PGD}}$ model is obtained in [52]. It achieves better robustness by using early stopping and a proper weight decay value. To fairly compare with this model, we reproduce the results of [52] and train ADT based models using the same settings/hyperparameters as in [52], where a WRN-34-10 model is adopted.

Table 6: Comparison with the state-of-the-art AT$_{\mathrm{PGD}}$ model in [52].

| Model | $\mathcal{A}_{\mathrm{nat}}$ | PGD-10 | PGD-20 | PGD-100 |
|---|---|---|---|---|
| AT$_{\mathrm{PGD}}$ | 86.41% | 55.90% | 54.52% | 54.20% |
| ADT$_{\mathrm{EXP}}$ | 86.49% | **56.84%** | **55.43%** | **55.01%** |
| ADT$_{\mathrm{EXP-AM}}$ | 87.27% | 56.28% | 54.88% | 54.58% |
| ADT$_{\mathrm{IMP-AM}}$ | **87.38%** | 56.63% | 55.10% | 54.43% |

The results of those models on CIFAR-10 are shown in Table 6. By using the same training settings, our models can also improve the performance over AT$_{\mathrm{PGD}}$.

## 6 Conclusion

In this paper, we introduced an adversarial distributional training framework for learning robust DNNs. ADT can learn an adversarial distribution to characterize heterogeneous adversarial examples around a natural one under an entropic regularizer. Through a theoretical analysis, we provided a general algorithm for solving ADT, and proposed to parameterize the adversarial distributions in ADT with three different approaches, ranging from the typical Gaussian distributions to the flexible implicit distributions. We conducted extensive experiments on CIFAR-10, CIFAR-100, and SVHN to demonstrate the effectiveness of ADT on building robust DNNs, compared with the state-of-the-art adversarial training methods.

## Broader Impact

The existence of adversarial examples poses potential security threats to machine learning models, when they are deployed to real-world applications, especially the security-sensitive ones, such as autonomous driving, healthcare, and finance. The model vulnerability to such small perturbations could lower the confidence of the public on machine learning techniques. Therefore, it is of particular importance to develop more robust models. This work is dedicated to developing a new learning framework to train robust deep learning models, which is the potential positive impact of this work in the society. Nevertheless, many works have shown that there is an inherent trade-off between robustness and natural accuracy [65, 78], that a classifier trained to be adversarially robust would introduce degraded accuracy on clean data, and our work is no exception. Although our proposed methods can obtain higher natural accuracy than the previous adversarial training methods, they still have lower natural accuracy than a standard trained model. The degeneration in natural accuracy could be a negative consequence. From a different perspective, adversarial examples also provide an opportunity to protect private information of users [45, 73]. Building a robust model could negatively impact users' ability to hide their privacy from the excessive unauthorized recognition systems.

## Acknowledgements

This work was supported by the National Key Research and Development Program of China (No.2020AAA0104304), NSFC Projects (Nos. 61620106010, 62076147, U19B2034, U1811461), Beijing Academy of Artificial Intelligence (BAAI), Tsinghua-Huawei Joint Research Program, a grant from Tsinghua Institute for Guo Qiang, Tiangong Institute for Intelligent Computing, and the NVIDIA NVAIL Program with GPU/DGX Acceleration. Yinpeng Dong was supported by MSRA and Baidu fellowships.

## Footnotes

[2]Code is available at `https://github.com/dongyp13/Adversarial-Distributional-Training`.

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
