[Supplementary Material]

# Supplementary Material for: Adversarial Distributional Training for Robust Deep Learning

**Yinpeng Dong**[*], **Zhijie Deng**[*], **Tianyu Pang, Jun Zhu, Hang Su**[†]
Dept. of Comp. Sci. & Tech., Institute for AI, BNRist Center
Tsinghua-Bosch Joint ML Center, THBI Lab, Tsinghua University, Beijing, 100084 China
{dyp17, dzj17, pty17}@mails.tsinghua.edu.cn, {suhangss, dcszj}@mail.tsinghua.edu.cn

## A  Technical details and algorithms

### A.1  ADT$_{\text{EXP}}$

We provide the algorithm for ADT$_{\text{EXP}}$ in Alg. 2.

### A.2  ADT$_{\text{EXP-AM}}$

By amortizing the explicit adversarial distributions, we can rewrite the minimax problem of ADT as

$$\min_{\boldsymbol{\theta}} \max_{\boldsymbol{\phi}} \frac{1}{n} \sum_{i=1}^{n} \Big\{ \mathbb{E}_{p_{\boldsymbol{\phi}}(\boldsymbol{\delta}_i|\mathbf{x}_i)} \big[ \mathcal{L}(f_{\boldsymbol{\theta}}(\mathbf{x}_i + \boldsymbol{\delta}_i), y_i) \big] + \lambda \mathcal{H}(p_{\boldsymbol{\phi}}(\boldsymbol{\delta}_i|\mathbf{x}_i)) \Big\}, \tag{A.1}$$

where $\boldsymbol{\theta}$ and $\boldsymbol{\phi}$ are the parameters of the DNN classifier and the generator, respectively. During training, we perform stochastic gradient descent and ascent on $\boldsymbol{\theta}$ and $\boldsymbol{\phi}$ simultaneously, to accomplish adversarial training. To enable the gradients flowing from $\boldsymbol{\delta}_i$ to $\boldsymbol{\phi}$, we apply the same reparameterization strategy as in Sec. 3.1. In practice, we only use one MC sample for each data. We provide the algorithm for ADT$_{\text{EXP-AM}}$ in Alg. 3.

### A.3  ADT$_{\text{IMP-AM}}$

For the implicit adversarial distributions, we have no access to the density $p_{\boldsymbol{\phi}}(\boldsymbol{\delta}_i|\mathbf{x}_i)$, such that the entropy of the adversarial distributions cannot be estimated exactly[1]. An appealing alternative is to maximize the variational lower bound of the entropy [2] for its simplicity and success in GANs [3]. In our case, for a natural input $\mathbf{x}_i$, we can similarly derive the following lower bound stemming from the mutual information between the perturbation $\boldsymbol{\delta}_i$ and the random noise $\mathbf{z}$ (proof in Appendix B.3) as

$$\mathcal{H}(p_{\boldsymbol{\phi}}(\boldsymbol{\delta}_i|\mathbf{x}_i)) \geq \mathcal{U}(q) = \mathbb{E}_{p(\mathbf{z})} \log q(\mathbf{z}|g_{\boldsymbol{\phi}}(\mathbf{z};\mathbf{x}_i)) + c, \tag{A.2}$$

where $c$ is a constant and $q(\cdot|\cdot)$ is an introduced variational distribution. In practice, we implement $q$ as a diagonal Gaussian, whose mean and standard derivation are given by a $\boldsymbol{\psi}$-parameterized neural network. Then we have the training objective as

$$\min_{\boldsymbol{\theta}} \max_{\boldsymbol{\phi},\boldsymbol{\psi}} \frac{1}{n} \sum_{i=1}^{n} \Big\{ \mathbb{E}_{p(\mathbf{z})} \big[ \mathcal{L}(f_{\boldsymbol{\theta}}(\mathbf{x}_i + g_{\boldsymbol{\phi}}(\mathbf{z};\mathbf{x}_i)), y_i) + \lambda \log q_{\boldsymbol{\psi}}(\mathbf{z}|g_{\boldsymbol{\phi}}(\mathbf{z};\mathbf{x}_i)) \big] \Big\}, \tag{A.3}$$

which is solved by simultaneous stochastic gradient descent and ascent on $\boldsymbol{\theta}$ and $(\boldsymbol{\phi},\boldsymbol{\psi})$. We provide the algorithm for ADT$_{\text{IMP-AM}}$ in Alg. 4.

---

[*]Equal contribution. † Corresponding author.
[1]We can also directly estimate the gradient of the entropy with advanced techniques such as spectral Stein gradient estimator [15], and we leave this for future work.

---
**Algorithm 2** The training algorithm for $\text{ADT}_{\text{EXP}}$
---
**Input:** Training data $\mathcal{D}$, objective function $\mathcal{J}\big(p_{\phi_i}(\boldsymbol{\delta}_i), \boldsymbol{\theta}\big)$, training epochs $N$, the number of inner maximization steps $T$, the number of MC samples for gradient estimation in each step $k$, and learning rates $\eta_{\boldsymbol{\theta}}, \eta_{\boldsymbol{\phi}}$.

1: Initialize $\boldsymbol{\theta}$;
2: **for** epoch $= 1$ **to** $N$ **do**
3:     **for** each minibatch $\mathcal{B} \subset \mathcal{D}$ **do**
4:         **for** each input $(\mathbf{x}_i, y_i) \in \mathcal{B}$ **do**
5:             Initialize $\boldsymbol{\phi}_i$;
6:             **for** $t = 1$ **to** $T$ **do**
7:                 Calculate the gradient $\mathbf{g}_i$ of $\boldsymbol{\phi}_i$ by Eq. (10) via MC integration using $k$ samples;
8:                 Update $\boldsymbol{\phi}_i$ with gradient ascent

$$\boldsymbol{\phi}_i \leftarrow \boldsymbol{\phi}_i + \eta_{\boldsymbol{\phi}} \cdot \mathbf{g}_i.$$

9:             **end for**
10:         **end for**
11:         Update $\boldsymbol{\theta}$ with stochastic gradient descent

$$\boldsymbol{\theta} \leftarrow \boldsymbol{\theta} - \eta_{\boldsymbol{\theta}} \cdot \mathbb{E}_{(\mathbf{x}_i, y_i) \in \mathcal{B}}\big[\nabla_{\boldsymbol{\theta}} \mathcal{J}\big(p_{\phi_i}(\boldsymbol{\delta}_i), \boldsymbol{\theta}\big)\big].$$

12:     **end for**
13: **end for**
---

---
**Algorithm 3** The training algorithm for $\text{ADT}_{\text{EXP-AM}}$
---
**Input:** Training data $\mathcal{D}$, objective function in Eq. (A.1), training epochs $N$, and learning rates $\eta_{\boldsymbol{\theta}}$, $\eta_{\boldsymbol{\phi}}$.

1: Initialize $\boldsymbol{\theta}$ and $\boldsymbol{\phi}$;
2: **for** epoch $= 1$ **to** $N$ **do**
3:     **for** each minibatch $\mathcal{B} \subset \mathcal{D}$ **do**
4:         Input $\mathbf{x}_i$ to the generator and obtain the distribution parameters $(\boldsymbol{\mu}_i, \boldsymbol{\sigma}_i)$ for each data $(\mathbf{x}_i, y_i) \in \mathcal{B}$;
5:         Sample one $\boldsymbol{\delta}_i$ from the distribution defined by Eq. (8) given $(\boldsymbol{\mu}_i, \boldsymbol{\sigma}_i)$ for each $(\mathbf{x}_i, y_i) \in \mathcal{B}$ to approximately calculate the gradient of Eq. (A.1) w.r.t. $\boldsymbol{\theta}$ and $\boldsymbol{\phi}$, and obtain $\mathbf{g}_{\boldsymbol{\theta}}$ and $\mathbf{g}_{\boldsymbol{\phi}}$;
6:         Update $\boldsymbol{\theta}$ by: $\boldsymbol{\theta} \leftarrow \boldsymbol{\theta} - \eta_{\boldsymbol{\theta}} \cdot \mathbf{g}_{\boldsymbol{\theta}}$.
7:         Update $\boldsymbol{\phi}$ by: $\boldsymbol{\phi} \leftarrow \boldsymbol{\phi} + \eta_{\boldsymbol{\phi}} \cdot \mathbf{g}_{\boldsymbol{\phi}}$.
8:     **end for**
9: **end for**
---

# B Proofs

We provide the proofs in this section.

## B.1 Proof of Theorem 1

**Assumption 1.** *The loss function $\mathcal{J}\big(p(\boldsymbol{\delta}_i), \boldsymbol{\theta}\big)$ is continuously differentiable w.r.t. $\boldsymbol{\theta}$.*

**Assumption 2.** *Probability density functions of distributions in $\mathcal{P}$ are bounded and equicontinuous.*

**Remark 1.** *For the explicit adversarial distributions defined in Eq. (8), we can assume that the mean and standard deviation of each dimension satisfy $|\boldsymbol{\mu}_i^{(j)}| < \kappa_{\mu}$ and $\kappa_{\sigma}^{lo} < \boldsymbol{\sigma}_i^{(j)} < \kappa_{\sigma}^{up}$, where $\kappa_{\mu}$, $\kappa_{\sigma}^{lo}$, and $\kappa_{\sigma}^{up}$ are constants. Note that they can be easily satisfied since we add an entropic regularization term into the training objective (5), such that the mean cannot be too large while the standard deviation cannot be too small or too large given Eq. (11). In practice, we can clip $\boldsymbol{\mu}_i^{(j)}$ and $\boldsymbol{\sigma}_i^{(j)}$ if they are out of the thresholds. Then we can prove that the density functions of the explicit adversarial distributions defined in Eq. (8) are bounded and equicontinuous, satisfying Assumption 2. However, for the implicit adversarial distributions introduced in Sec. 3.3, we cannot prove that Assumption 2 is satisfied. Though unsatisfied, the experiments suggest that we can still rely on Theorem 1 and the general algorithm for training.*

---

**Algorithm 4** The training algorithm for ADT$_{\text{IMP-AM}}$

---

**Input:** Training data $\mathcal{D}$, objective function in Eq. (A.3), training epochs $N$, and learning rates $\eta_{\boldsymbol{\theta}}$, $\eta_{\boldsymbol{\phi}}$, $\eta_{\boldsymbol{\psi}}$.

1: Initialize $\boldsymbol{\theta}$, $\boldsymbol{\phi}$, and $\boldsymbol{\psi}$;
2: **for** epoch = 1 **to** $N$ **do**
3:     **for** each minibatch $\mathcal{B} \subset \mathcal{D}$ **do**
4:         For each $(\mathbf{x}_i, y_i) \in \mathcal{B}$, sample a noise $\mathbf{z}_i$ from $\text{U}(-1,1)$.
5:         Use the sampled noises to approximately calculate the gradient of Eq. (A.3) w.r.t. $\boldsymbol{\theta}$, $\boldsymbol{\phi}$, and $\boldsymbol{\psi}$, and obtain $\mathbf{g}_{\boldsymbol{\theta}}$, $\mathbf{g}_{\boldsymbol{\phi}}$, and $\mathbf{g}_{\boldsymbol{\psi}}$.
6:         Update $\boldsymbol{\theta}$ by: $\boldsymbol{\theta} \leftarrow \boldsymbol{\theta} - \eta_{\boldsymbol{\theta}} \cdot \mathbf{g}_{\boldsymbol{\theta}}$.
7:         Update $\boldsymbol{\phi}$ by: $\boldsymbol{\phi} \leftarrow \boldsymbol{\phi} + \eta_{\boldsymbol{\phi}} \cdot \mathbf{g}_{\boldsymbol{\phi}}$.
8:         Update $\boldsymbol{\psi}$ by: $\boldsymbol{\psi} \leftarrow \boldsymbol{\psi} + \eta_{\boldsymbol{\psi}} \cdot \mathbf{g}_{\boldsymbol{\psi}}$.
9:     **end for**
10: **end for**

---

*Proof.* Due to the diagonal covariance matrix, each dimension of $p_{\boldsymbol{\phi}_i}(\boldsymbol{\delta}_i)$ is independent. Thus we only consider one dimension of $\boldsymbol{\delta}_i$. For clarity, we denote $\boldsymbol{\mu}_i^{(j)}$, $\boldsymbol{\sigma}_i^{(j)}$, $\mathbf{r}^{(j)}$, $\mathbf{u}_i^{(j)}$, and $\boldsymbol{\delta}_i^{(j)}$ as $\mu$, $\sigma$, $r$, $u$, and $\delta$, respectively. The probability density function of $\delta$ is (see Appendix B.2 for details)

$$
\begin{aligned}
p(\delta) =& \frac{1}{\sqrt{2\pi}\sigma} \exp\left( -\frac{(\frac{1}{2}\log\frac{\epsilon+\delta}{\epsilon-\delta} - \mu)^2}{2\sigma^2} \right) \cdot \frac{\epsilon}{\epsilon^2 - \delta^2} \\
=& \frac{1}{\sqrt{2\pi}\sigma} \exp\left( -\frac{r^2}{2} \right) \cdot \frac{1}{1 - \tanh(\mu+\sigma r)^2} \cdot \frac{1}{\epsilon}.
\end{aligned}
$$

By calculation, we have

$$
\begin{aligned}
p(\delta) =& \frac{1}{4\sqrt{2\pi}\sigma\epsilon} \left[ \exp\left( -\frac{r^2}{2} + 2\sigma r + 2\mu \right) + 2\exp\left( -\frac{r^2}{2} \right) + \exp\left( -\frac{r^2}{2} - 2\sigma r - 2\mu \right) \right] \\
\leq& \frac{1}{4\sqrt{2\pi}\sigma\epsilon} \left[ \exp\left( 2\sigma^2 + 2\mu \right) + 2 + \exp\left( 2\sigma^2 - 2\mu \right) \right] \\
\leq& \frac{1}{4\sqrt{2\pi}\kappa_\sigma^{lo}\epsilon} \left[ 2\exp\left( 2(\kappa_\sigma^{up})^2 + 2\kappa_\mu \right) + 2 \right].
\end{aligned}
$$

Hence, $p(\delta)$ is bounded. And the probability density function $p_{\boldsymbol{\phi}_i}(\boldsymbol{\delta}_i)$ is also bounded since it equals to the product of $p(\delta)$ across all dimensions.

We next prove $p(\delta)$ is Lipschitz continuous at $\delta \in (-\epsilon, \epsilon)$. By calculating the derivative of $p(\delta)$, we have

$$
\begin{aligned}
p'(\delta) =& \frac{1}{\sqrt{2\pi}\sigma} \exp\left( -\frac{(\frac{1}{2}\log\frac{\epsilon+\delta}{\epsilon-\delta} - \mu)^2}{2\sigma^2} \right) \cdot \left[ \frac{2\epsilon\delta}{(\epsilon^2-\delta^2)^2} + \frac{\frac{1}{2}\log\frac{\epsilon+\delta}{\epsilon-\delta} - \mu}{\sigma^2} \cdot \left( \frac{\epsilon}{\epsilon^2-\delta^2} \right)^2 \right] \\
=& \frac{1}{\sqrt{2\pi}\sigma} \exp\left( -\frac{r^2}{2} \right) \cdot \left[ \frac{2\tanh(\mu+\sigma r)}{\epsilon^2(1-\tanh(\mu+\sigma r)^2)^2} + \frac{r}{\sigma\epsilon^2(1-\tanh(\mu+\sigma r)^2)^2} \right].
\end{aligned}
$$

Note that although $p'(\delta)$ has a more complicated form, the quadratic term inside $\exp$ is still $-\frac{r^2}{2}$. Hence, $p'(\delta)$ can also be bounded by a constant. Then $p(\delta)$ as well as $p_{\boldsymbol{\phi}_i}(\boldsymbol{\delta}_i)$ are Lipschitz continuous. The Lipschitz constant only concerns with $\epsilon$, $\kappa_\mu$, $\kappa_\sigma^{lo}$, and $\kappa_\sigma^{up}$. Hence, the set of explicit distributions in $\mathcal{P}$ with a common Lipschitz constant is equicontinuous.

Combining the results, we prove that the probability density functions of the set of explicit adversarial distributions defined in Eq. (8) are bound and equicontinuous, which satisfies Assumption 2. $\square$

**Remark 2.** *Assumption 2 is used to make the search space $\mathcal{P}$ of the inner problem in ADT compact, as can be seen in Lemma 1. However, it is a sufficient bot not necessary condition of making $\mathcal{P}$ compact. For example, if $\mathcal{P}$ only contains Delta distributions, ADT degenerates to the AT formulation in Eq. (1) and $\mathcal{P}$ can be represented by $\mathcal{S}$. In this case, it is easy to see that Assumption 2 is not satisfied but the search space of the inner problem is also compact.*

**Theorem 1.** *Suppose Assumptions 1 and 2 hold. We define $\rho(\boldsymbol{\theta}) = \max_{p(\boldsymbol{\delta}_i) \in \mathcal{P}} \mathcal{J}(p(\boldsymbol{\delta}_i), \boldsymbol{\theta})$, and $\mathcal{P}^*(\boldsymbol{\theta}) = \{p(\boldsymbol{\delta}_i) \in \mathcal{P} : \mathcal{J}(p(\boldsymbol{\delta}_i), \boldsymbol{\theta}) = \rho(\boldsymbol{\theta})\}$. Then $\rho(\boldsymbol{\theta})$ is directionally differentiable, and its directional derivative along the direction $\mathbf{v}$ satisfies*

$$\rho'(\boldsymbol{\theta}; \mathbf{v}) = \sup_{p(\boldsymbol{\delta}_i) \in \mathcal{P}^*(\boldsymbol{\theta})} \mathbf{v}^\top \nabla_{\boldsymbol{\theta}} \mathcal{J}(p(\boldsymbol{\delta}_i), \boldsymbol{\theta}). \tag{B.1}$$

*Particularly, when $\mathcal{P}^*(\boldsymbol{\theta}) = \{p^*(\boldsymbol{\delta}_i)\}$ only contains one maximizer, $\rho(\boldsymbol{\theta})$ is differentiable at $\boldsymbol{\theta}$ and*

$$\nabla_{\boldsymbol{\theta}} \rho(\boldsymbol{\theta}) = \nabla_{\boldsymbol{\theta}} \mathcal{J}(p^*(\boldsymbol{\delta}_i), \boldsymbol{\theta}). \tag{B.2}$$

*Proof.* Recall that $\mathcal{P}$ is a set of distributions, which can be expressed by their probability density functions. The support of these functions is contained in $\mathcal{S}$ and these functions are equicontinuous by Assumption 2. $\mathcal{S} = \{\boldsymbol{\delta} : \|\boldsymbol{\delta}\|_\infty \leq \epsilon\}$ is the allowed perturbation set. The Euclidean distance $\ell_2$ defines a metric on $\mathcal{S}$. We let

$$\mathcal{C}(\mathcal{S}, \mathbb{R}) = \{h : \mathcal{S} \to \mathbb{R} | h \text{ is continuous}\}$$

be the collection of all continuous functions from $\mathcal{S}$ to $\mathbb{R}$. Then $\mathcal{P}$ is a subset of $\mathcal{C}(\mathcal{S}, \mathbb{R})$. We let

$$d_{\mathcal{C}}(p, q) = \max_{\boldsymbol{\delta} \in \mathcal{S}} |p(\boldsymbol{\delta}) - q(\boldsymbol{\delta})|$$

for all $p, q \in \mathcal{C}(\mathcal{S}, \mathbb{R})$ be a metric on $\mathcal{C}(\mathcal{S}, \mathbb{R})$. Then we can see that $(\mathcal{C}(\mathcal{S}, \mathbb{R}), d_{\mathcal{C}})$ is a metric space.

We state the following lemma to prove that $\mathcal{P}$ is compact.

**Lemma 1.** *(Arzelà-Ascoli's Theorem) Let $(X, d_X)$ be a compact metric space. A subset $\mathcal{K}$ of $\mathcal{C}(X, \mathbb{R})$ is compact if and only if it is closed, bounded, and equicontinuous.*

Since $(\mathcal{S}, \ell_2)$ is a compact metric space, and $\mathcal{P}$ is closed, bounded, and equicontinuous given by Assumption 2, we can see that $\mathcal{P}$ is compact by Lemma 1.

We next need to prove that the loss function $\mathcal{J}(p(\boldsymbol{\delta}_i), \boldsymbol{\theta})$ is continuously differentiable w.r.t. both $p(\boldsymbol{\delta}_i)$ and $\boldsymbol{\theta}$, i.e., the gradient $\nabla_{\boldsymbol{\theta}} \mathcal{J}(p(\boldsymbol{\delta}_i), \boldsymbol{\theta})$ is joint continuous on $\mathcal{P} \times \mathbb{R}^m$, where $m$ is the dimension of $\theta$.

To prove it, we first define a new metric on $\mathcal{P} \times \mathbb{R}^m$ as

$$d_{mix}((p_1, \boldsymbol{\theta}_1), (p_2, \boldsymbol{\theta}_2)) = d_{\mathcal{C}}(p_1, p_2) + \ell_2(\boldsymbol{\theta}_1, \boldsymbol{\theta}_2).$$

Then $(\mathcal{P} \times \mathbb{R}^m, d_{mix})$ is a new metric space.

By definition, given a point $(p_0, \boldsymbol{\theta}_0) \in \mathcal{P} \times \mathbb{R}^m$, if for each $\tau > 0$, there is a $\gamma > 0$, such that

$$\ell_2(\nabla_{\boldsymbol{\theta}} \mathcal{J}(p(\boldsymbol{\delta}_i), \boldsymbol{\theta}), \nabla_{\boldsymbol{\theta}} \mathcal{J}(p_0(\boldsymbol{\delta}_i), \boldsymbol{\theta}_0)) < \tau$$

whenever $d_{mix}((p, \boldsymbol{\theta}), (p_0, \boldsymbol{\theta}_0)) < \gamma$, then the function $\nabla_{\boldsymbol{\theta}} \mathcal{J}(p(\boldsymbol{\delta}_i), \boldsymbol{\theta})$ is continuous at $(p_0, \boldsymbol{\theta}_0)$. If for all points in $\mathcal{P} \times \mathbb{R}^m$, the function is continuous, then $\nabla_{\boldsymbol{\theta}} \mathcal{J}(p(\boldsymbol{\delta}_i), \boldsymbol{\theta})$ is continuous on $\mathcal{P} \times \mathbb{R}^m$.

To show that, we first have

$$\ell_2(\nabla_{\boldsymbol{\theta}} \mathcal{J}(p(\boldsymbol{\delta}_i), \boldsymbol{\theta}), \nabla_{\boldsymbol{\theta}} \mathcal{J}(p_0(\boldsymbol{\delta}_i), \boldsymbol{\theta}_0))$$
$$\leq \ell_2(\nabla_{\boldsymbol{\theta}} \mathcal{J}(p(\boldsymbol{\delta}_i), \boldsymbol{\theta}), \nabla_{\boldsymbol{\theta}} \mathcal{J}(p(\boldsymbol{\delta}_i), \boldsymbol{\theta}_0)) + \ell_2(\nabla_{\boldsymbol{\theta}} \mathcal{J}(p(\boldsymbol{\delta}_i), \boldsymbol{\theta}_0), \nabla_{\boldsymbol{\theta}} \mathcal{J}(p_0(\boldsymbol{\delta}_i), \boldsymbol{\theta}_0)). \tag{B.3}$$

We already have that the loss function $\mathcal{J}(p(\boldsymbol{\delta}_i), \boldsymbol{\theta})$ is continuously differentiable w.r.t. $\boldsymbol{\theta}$ by Assumption 1. Then given $\frac{\tau}{2}$, there is a $\gamma_1$, such that

$$\ell_2(\nabla_{\boldsymbol{\theta}} \mathcal{J}(p(\boldsymbol{\delta}_i), \boldsymbol{\theta}), \nabla_{\boldsymbol{\theta}} \mathcal{J}(p(\boldsymbol{\delta}_i), \boldsymbol{\theta}_0)) < \frac{\tau}{2}$$

whenever $\ell_2(\boldsymbol{\theta}, \boldsymbol{\theta}_0) < \gamma_1$.

For the second term of the RHS of Eq. (B.3), we have

$$\ell_2(\nabla_{\boldsymbol{\theta}} \mathcal{J}(p(\boldsymbol{\delta}_i), \boldsymbol{\theta}_0), \nabla_{\boldsymbol{\theta}} \mathcal{J}(p_0(\boldsymbol{\delta}_i), \boldsymbol{\theta}_0)) = \left\| \nabla_{\boldsymbol{\theta}} (\mathcal{J}(p(\boldsymbol{\delta}_i), \boldsymbol{\theta}_0) - \mathcal{J}(p_0(\boldsymbol{\delta}_i), \boldsymbol{\theta}_0)) \right\|_2$$
$$= \left\| \int_{\mathcal{S}} (p(\boldsymbol{\delta}_i) - p_0(\boldsymbol{\delta}_i)) \nabla_{\boldsymbol{\theta}} \mathcal{L}(f_{\boldsymbol{\theta}}(\mathbf{x}_i + \boldsymbol{\delta}_i), y_i) d\boldsymbol{\delta}_i \right\|_2$$
$$\leq d_{\mathcal{C}}(p, p_0) \cdot \int_{\mathcal{S}} \left\| \nabla_{\boldsymbol{\theta}} \mathcal{L}(f_{\boldsymbol{\theta}}(\mathbf{x}_i + \boldsymbol{\delta}_i), y_i) \right\|_2 d\boldsymbol{\delta}_i.$$

Therefore, for the given $\frac{\tau}{2}$, there is also a $\gamma_2$ which equals to

$$\gamma_2 = \frac{\tau}{2\int_{\mathcal{S}}\left\|\nabla_{\boldsymbol{\theta}}\mathcal{L}(f_{\boldsymbol{\theta}}(\mathbf{x}_i + \boldsymbol{\delta}_i), y_i)\right\|_2 d\boldsymbol{\delta}_i},$$

such that

$$\ell_2\big(\nabla_{\boldsymbol{\theta}}\mathcal{J}\big(p(\boldsymbol{\delta}_i), \boldsymbol{\theta}_0\big), \nabla_{\boldsymbol{\theta}}\mathcal{J}\big(p_0(\boldsymbol{\delta}_i), \boldsymbol{\theta}_0\big)\big) < \frac{\tau}{2}$$

whenever $d_{\mathcal{C}}(p, p_0) < \gamma_2$.

Combining the results, for a given $\tau > 0$, we can set $\gamma = \gamma_1 + \gamma_2$, such that

$$\ell_2\big(\nabla_{\boldsymbol{\theta}}\mathcal{J}\big(p(\boldsymbol{\delta}_i), \boldsymbol{\theta}\big), \nabla_{\boldsymbol{\theta}}\mathcal{J}\big(p_0(\boldsymbol{\delta}_i), \boldsymbol{\theta}_0\big)\big) < \tau$$

whenever $d_{mix}((p, \boldsymbol{\theta}), (p_0, \boldsymbol{\theta}_0)) < \gamma$. Thus we have proven that the loss function $\mathcal{J}\big(p(\boldsymbol{\delta}_i), \boldsymbol{\theta}\big)$ is continuously differentiable w.r.t. both $p(\boldsymbol{\delta}_i)$ and $\boldsymbol{\theta}$.

Given the above results, we can directly apply Danskin's theorem [4] to prove Theorem 1. We state the Danskin's theorem in the following lemma.

**Lemma 2.** *(**Danskin's Theorem**) Let $\mathcal{Q}$ be a nonempty compact topological space and $h : \mathcal{Q} \times \mathbb{R}^m \to \mathbb{R}$ be a function satisfying that $h(q, \cdot)$ is differentiable for every $q \in \mathcal{Q}$ and $\nabla_{\boldsymbol{\theta}} h(q, \boldsymbol{\theta})$ is continuous on $\mathcal{Q} \times \mathbb{R}^m$. We define $\Psi(\boldsymbol{\theta}) = \max_{q \in \mathcal{Q}} h(q, \boldsymbol{\theta})$, and $\mathcal{Q}^*(\boldsymbol{\theta}) = \{q \in \mathcal{Q} : h(q, \boldsymbol{\theta}) = \Psi(\boldsymbol{\theta})\}$. Then $\Psi(\boldsymbol{\theta})$ is directionally differentiable, and its directional derivative along the direction $\mathbf{v}$ satisfies*

$$\Psi'(\boldsymbol{\theta}; \mathbf{v}) = \sup_{q \in \mathcal{Q}^*(\boldsymbol{\theta})} \mathbf{v}^\top \nabla_{\boldsymbol{\theta}} h(q, \boldsymbol{\theta}).$$

*Particularly, when $\mathcal{Q}^*(\boldsymbol{\theta}) = \{q^*\}$ only contains one maximizer, $\Psi(\boldsymbol{\theta})$ is differentiable at $\boldsymbol{\theta}$ and*

$$\nabla_{\boldsymbol{\theta}} \Psi(\boldsymbol{\theta}) = \nabla_{\boldsymbol{\theta}} h(q^*, \boldsymbol{\theta}).$$

If we let $\mathcal{Q} = \mathcal{P}$ and $h = \mathcal{J}$ in Lemma 2, we can directly prove Theorem 1. $\qquad\square$

## B.2 Proof of Eq. (11)

The variable $\boldsymbol{\delta}_i$ has the following sampling process

$$\boldsymbol{\delta}_i = \epsilon \cdot \tanh(\mathbf{u}_i), \quad \mathbf{u}_i \sim \mathcal{N}(\boldsymbol{\mu}_i, \mathrm{diag}(\boldsymbol{\sigma}_i^2)),$$

whose negative log density is

$$\sum_{j=1}^d \big(\frac{1}{2}(\mathbf{r}^{(j)})^2 + \frac{\log 2\pi}{2} + \log \boldsymbol{\sigma}_i^{(j)} + \log(1 - \tanh(\boldsymbol{\mu}_i^{(j)} + \boldsymbol{\sigma}_i^{(j)}\mathbf{r}^{(j)})^2) + \log \epsilon\big),$$

where the superscript $j$ denotes the $j$-th element of a vector.

*Proof.* Due to the usage of the diagonal covariance matrix, each dimension in the sampled perturbation $\boldsymbol{\delta}_i$ is independent. Thus we can simply calculate the negative log density in each dimension of $\boldsymbol{\delta}_i$. For clarity, we also denote $\boldsymbol{\mu}_i^{(j)}, \boldsymbol{\sigma}_i^{(j)}, \mathbf{r}^{(j)}, \mathbf{u}_i^{(j)}$, and $\boldsymbol{\delta}_i^{(j)}$ as $\mu, \sigma, r, u$, and $\delta$, respectively. Based on the sampling procedure in Eq. (8), we have $\delta = \epsilon \cdot \tanh(u)$ and $u = \mu + \sigma r$.

Note that $r$ has density: $p(r) = \frac{1}{\sqrt{2\pi}} \exp\left(-\frac{r^2}{2}\right)$. Apply the *transformation of variable* approach, we have the density of $u$ as

$$p(u) = \frac{1}{\sqrt{2\pi}} \exp\left(-\frac{r^2}{2}\right) \cdot \left|\frac{d}{du}\left(\frac{u - \mu}{\sigma}\right)\right| = \frac{1}{\sqrt{2\pi}\sigma} \exp\left(-\frac{r^2}{2}\right).$$

Let $\beta = \tanh(u)$, then the inverse transformation is $u = \tanh^{-1}(\beta) = \frac{1}{2}\log(\frac{1+\beta}{1-\beta})$, whose derivative w.r.t. $\beta$ is $\frac{1}{1-\beta^2}$.

Then, by applying the *transformation of variable* approach again, we have the density of $\beta$ as

$$p(\beta) = \frac{1}{\sqrt{2\pi}\sigma} \exp\left(-\frac{r^2}{2}\right) \cdot \frac{1}{1-\beta^2} = \frac{1}{\sqrt{2\pi}\sigma} \exp\left(-\frac{r^2}{2}\right) \cdot \frac{1}{1 - \tanh(\mu + \sigma r)^2}.$$

Therefore, the density of $\delta$ which equals to $\epsilon \cdot \beta$ can be derived similarly, and eventually we obtain

$$p(\delta) = \frac{1}{\sqrt{2\pi}\sigma} \exp\left(-\frac{r^2}{2}\right) \cdot \frac{1}{1 - \tanh(\mu + \sigma r)^2} \cdot \frac{1}{\epsilon}.$$

Consequently, the negative log density of $p(\delta)$ is

$$-\log p(\delta) = \frac{r^2}{2} + \frac{\log 2\pi}{2} + \log \sigma + \log(1 - \tanh(\mu + \sigma r)^2) + \log \epsilon.$$

Sum over all of the dimensions and we complete the proof of Eq. (11). $\qquad\square$

### B.3 Proof of Eq. (A.2)

Given an example $\mathbf{x}_i$, we define an implicit adversarial distribution $p_\phi(\boldsymbol{\delta}_i|\mathbf{x}_i)$ in the form of $\boldsymbol{\delta}_i = g_\phi(\mathbf{z}; \mathbf{x}_i), \mathbf{z} \sim p(\mathbf{z})$, where $g_\phi$ denotes a $\phi$-parameterized generator network. Then we can maximize the following variational lower bound to maximize the entropy of $p_\phi(\boldsymbol{\delta}_i|\mathbf{x}_i)$, as

$$\mathcal{H}(p_\phi(\boldsymbol{\delta}_i|\mathbf{x}_i)) \geq \mathcal{U}(q) = \mathbb{E}_{p(\mathbf{z})} \log q(\mathbf{z}|g_\phi(\mathbf{z}; \mathbf{x}_i)) + c$$

where $c$ is a constant and $q(\cdot|\cdot)$ is an introduced variational distribution.

*Proof.* We mainly follow [2] to provide the proof. Typically, we can view the Dirac generation distribution $p_\phi(\boldsymbol{\delta}_i|\mathbf{x}_i, \mathbf{z})$ as a peaked Gaussian with a fixed, diagonal covariance, then it will have a constant entropy. Considering $\mathbf{x}_i$ as a given condition, we can simply rewrite the generation distribution as $p_{\phi,i}(\boldsymbol{\delta}_i|\mathbf{z})$. Then we can define the joint distribution over $\boldsymbol{\delta}_i$ and $\mathbf{z}$ as $p_{\phi,i}(\boldsymbol{\delta}_i, \mathbf{z}) = p_{\phi,i}(\boldsymbol{\delta}_i|\mathbf{z})p_{\phi,i}(\mathbf{z})$. $p_{\phi,i}(\mathbf{z}) = p(\mathbf{z})$ is simply a predefined prior with a constant entropy. Then, we can further define the marginal $p_{\phi,i}(\boldsymbol{\delta}_i)$ whose entropy is of our interest and the posterior $p_{\phi,i}(\mathbf{z}|\boldsymbol{\delta}_i)$. Consider the mutual information between $\boldsymbol{\delta}_i$ and $\mathbf{z}$

$$\mathcal{I}(p_{\phi,i}(\boldsymbol{\delta}_i); p_{\phi,i}(\mathbf{z})) = \mathcal{H}(p_{\phi,i}(\boldsymbol{\delta}_i)) - \mathcal{H}(p_{\phi,i}(\boldsymbol{\delta}_i|\mathbf{z})) = \mathcal{H}(p_{\phi,i}(\mathbf{z})) - \mathcal{H}(p_{\phi,i}(\mathbf{z}|\boldsymbol{\delta}_i)).$$

Thus, we can calculate the entropy of $\boldsymbol{\delta}_i$ as

$$\mathcal{H}(p_{\phi,i}(\boldsymbol{\delta}_i)) = \mathcal{H}(p_{\phi,i}(\mathbf{z})) - \mathcal{H}(p_{\phi,i}(\mathbf{z}|\boldsymbol{\delta}_i)) + \mathcal{H}(p_{\phi,i}(\boldsymbol{\delta}_i|\mathbf{z})).$$

As stated, the first term and the last term are constant w.r.t. the parameter $\phi$. Therefore, maximizing $\mathcal{H}(p_{\phi,i}(\boldsymbol{\delta}_i))$ corresponds to maximizing the negative conditional entropy

$$-\mathcal{H}(p_{\phi,i}(\mathbf{z}|\boldsymbol{\delta}_i)) = \mathbb{E}_{\boldsymbol{\delta}_i \sim p_{\phi,i}(\boldsymbol{\delta}_i)} \left[ \mathbb{E}_{\mathbf{z} \sim p_{\phi,i}(\mathbf{z}|\boldsymbol{\delta}_i)} [\log p_{\phi,i}(\mathbf{z}|\boldsymbol{\delta}_i)] \right].$$

We still cannot optimize this as we have no access to the posterior. As an alternative, we resort to the variational inference technique to tackle this problem. We introduce a variational distribution $q(\mathbf{z}|\boldsymbol{\delta}_i)$ to approximate the true posterior, and derive the following lower bound

$$
\begin{aligned}
-\mathcal{H}(p_{\phi,i}(\mathbf{z}|\boldsymbol{\delta}_i)) &= \mathbb{E}_{\boldsymbol{\delta}_i \sim p_{\phi,i}(\boldsymbol{\delta}_i)} \left[ \mathbb{E}_{\mathbf{z} \sim p_{\phi,i}(\mathbf{z}|\boldsymbol{\delta}_i)} [\log q(\mathbf{z}|\boldsymbol{\delta}_i)] \right] + \mathcal{D}_{KL}(p_{\phi,i}(\mathbf{z}|\boldsymbol{\delta}_i)||q(\mathbf{z}|\boldsymbol{\delta}_i)) \\
&\geq \mathbb{E}_{\boldsymbol{\delta}_i \sim p_{\phi,i}(\boldsymbol{\delta}_i)} \left[ \mathbb{E}_{\mathbf{z} \sim p_{\phi,i}(\mathbf{z}|\boldsymbol{\delta}_i)} [\log q(\mathbf{z}|\boldsymbol{\delta}_i)] \right] \\
&= \mathbb{E}_{\mathbf{z}, \boldsymbol{\delta}_i \sim p_{\phi,i}(\mathbf{z}, \boldsymbol{\delta}_i)} [\log q(\mathbf{z}|\boldsymbol{\delta}_i)] \\
&= \underbrace{\mathbb{E}_{\mathbf{z} \sim p_{\phi,i}(\mathbf{z})} \left[ \mathbb{E}_{\boldsymbol{\delta}_i \sim p_{\phi,i}(\boldsymbol{\delta}_i|\mathbf{z})} [\log q(\mathbf{z}|\boldsymbol{\delta}_i)] \right]}_{\mathcal{U}'(q)},
\end{aligned}
$$

where $\mathcal{D}_{KL}$ represents the Kullback–Leibler divergence between distributions. Note that $p_{\phi,i}(\mathbf{z}) = p(\mathbf{z})$ is a prior and $p_{\phi,i}(\boldsymbol{\delta}_i|\mathbf{z}) = p_\phi(\boldsymbol{\delta}_i|\mathbf{x}_i, \mathbf{z})$ is Dirac distribution located at $\boldsymbol{\delta}_i = g_\phi(\mathbf{z}; \mathbf{x}_i)$. Thus, we can write the lower bound of the entropy $\mathcal{U}(q)$ as

$$\mathcal{H}(p_\phi(\boldsymbol{\delta}_i|\mathbf{x}_i)) \geq \mathcal{U}(q) = \mathcal{U}'(q) + c = \mathbb{E}_{\mathbf{z} \sim p(\mathbf{z})} \log q(\mathbf{z}|g_\phi(\mathbf{z}; \mathbf{x}_i)) + c,$$

which can be optimized effectively via Monte Carlo integration and standard back-propagation. Then we finish the proof of Eq. (A.2). $\qquad\square$

## C   Detailed experimental settings

We provide the detailed experimental settings in this section. All of the experiments are conducted on NVIDIA 2080 Ti GPUs. The source code of ADT is available at https://github.com/dongyp13/Adversarial-Distributional-Training.

Table 7: The network architectures used for the generators.

| In ADT$_{\text{EXP-AM}}$ | In ADT$_{\text{IMP-AM}}$ |
|---|---|
| input | **z** |
| $256 \times 3 \times 3$ conv | 256-dim fc layer |
| Residual block, 512 filters | 1024-dim fc layer |
| Residual block, 512 filters | reshape to $1 \times 32 \times 32$ |
| Residual block, 512 filters | concat with input |
| $6 \times 3 \times 3$ conv | $256 \times 3 \times 3$ conv |
| | Residual block, 512 filters |
| | Residual block, 512 filters |
| | Residual block, 512 filters |
| | $3 \times 3 \times 3$ conv |

Table 8: The network architecture used for instantiating the variational distribution $q$ in ADT$_{\text{IMP-AM}}$.

| Layers |
|---|
| input |
| $32 \times 5 \times 5$, stride 1 |
| $64 \times 4 \times 4$, stride 2 |
| $128 \times 4 \times 4$, stride 1 |
| $256 \times 4 \times 4$, stride 2 |
| Global average pooling |
| $128 \times 1 \times 1$, stride 1 |

## C.1 Datasets

We choose the CIFAR-10 [10], CIFAR-100 [10], and SVHN [14] datasets to conduct the experiments. CIFAR consists of a training set of $50,000$ and a test set of $10,000$ color images of resolution $32 \times 32$ with 10 classes in CIFAR-10 and 100 classes in CIFAR-100. SVHN is a 10-class house number classification dataset with $73,257$ training images and $26,032$ test images. During training, we perform standard data augmentation (i.e., horizontal flips and random crops from images with 4 pixels padded on each side) on CIFAR-10 and CIFAR-100, and use no data augmentation on SVHN. We do not use any data augmentation during testing.

## C.2 Network architectures

For the generator network in ADT$_{\text{EXP-AM}}$ and ADT$_{\text{IMP-AM}}$, we adopt a popular image-to-image architecture which has shown promise in neural style transfer and super-resolution [7, 18]. The network contains 3 residual blocks [5], with two extra convolutions at the beginning and the end. All convolutions in the generator have stride 1, and are immediately followed by batch normalization [6] and ReLU activation.

As found by [1], taking only the natural images as inputs to the generator network can lead to poor results. And they suggest to input the classifier's gradients as well. Based on this finding, we calculate the gradient of the loss function at the natural input $\mathbf{g}_i^1 = \nabla_{\mathbf{x}} \mathcal{L}(f_{\boldsymbol{\theta}}(\mathbf{x}_i), y_i)$, as well as the gradient of the loss function at the FGSM adversarial example $\mathbf{g}_i^2 = \nabla_{\mathbf{x}} \mathcal{L}(f_{\boldsymbol{\theta}}(\mathbf{x}_i + \boldsymbol{\delta}_i^{\text{FGSM}}), y_i)$, where $\boldsymbol{\delta}_i^{\text{FGSM}} = \epsilon \cdot \text{sign}(\nabla_{\mathbf{x}} \mathcal{L}(f_{\boldsymbol{\theta}}(\mathbf{x}_i), y_i))$, and then input $[\mathbf{x}_i, \mathbf{g}_i^1, \mathbf{g}_i^2]$ to the generator network.

In ADT$_{\text{EXP-AM}}$, the generator has 6 output channels to deliver the parameters (i.e., mean and standard derivation) of the explicit adversarial distributions. In ADT$_{\text{IMP-AM}}$, for each input we sample a 64-dim i.i.d. $\mathbf{z}$ from a uniform distribution $\text{U}(-1, 1)$, which is encoded with 2 fully connected (FC) layers and then fed into the generator along with the input image and gradients.

We elaborate the architectures of the generator networks in Table 7, and the architecture of $q$ in ADT$_{\text{IMP-AM}}$ in Table 8. In these tables, "C×H×W" means a convolutional layer with C filters size H×W, which is followed by batch normalization [6] and a ReLU nonlinearity (or LeakyReLU for layers in Table 8), except the last layers in the architectures. We use the residual block design in [5], which is composed of two $3 \times 3$ convolutions and a residual connection.

Table 9: Classification accuracy of the three proposed methods and baselines on CIFAR-100 and SVHN under white-box attacks. We mark the best results for each attack and the overall results that outperform the baselines in **bold**, and the overall best result in **blue**.

| Model | $\mathcal{A}_{\mathrm{nat}}$ | FGSM | PGD-20 | PGD-100 | MIM | C&W | FeaAttack | $\mathcal{A}_{\mathrm{rob}}$ |
|---|---|---|---|---|---|---|---|---|
| CIFAR-100, $\epsilon = 8/255$ | | | | | | | | |
| Standard | **78.59%** | 8.73% | 0.02% | 0.01% | 0.02% | 0.00% | 0.00% | 0.00% |
| $\mathrm{AT_{PGD}}$ | 61.45% | 30.78% | 25.71% | 25.40% | 26.60% | 25.80% | 33.95% | 24.49% |
| $\mathrm{ADT_{EXP}}$ | 62.70% | 34.22% | 28.96% | **28.60%** | **29.83%** | **28.99%** | **35.07%** | **27.13%** |
| $\mathrm{ADT_{EXP\text{-}AM}}$ | 62.84% | 36.28% | 29.01% | 28.46% | 29.68% | 28.78% | 34.91% | **26.87%** |
| $\mathrm{ADT_{IMP\text{-}AM}}$ | 64.07% | **39.39%** | **29.40%** | 28.43% | 29.64% | 28.76% | 35.00% | **26.80%** |
| SVHN, $\epsilon = 4/255$ | | | | | | | | |
| Standard | **96.12%** | 39.05% | 3.64% | 2.95% | 4.08% | 3.91% | 2.14% | 2.14% |
| $\mathrm{AT_{PGD}}$ | 95.07% | 82.19% | 74.22% | 73.79% | 74.56% | 74.77% | 73.51% | 73.38% |
| $\mathrm{ADT_{EXP}}$ | 95.70% | 86.72% | **77.01%** | **76.62%** | **77.18%** | **77.50%** | **75.64%** | **75.55%** |
| $\mathrm{ADT_{EXP\text{-}AM}}$ | 95.67% | 85.24% | 76.12% | 75.58% | 76.63% | 76.70% | 75.20% | **75.00%** |
| $\mathrm{ADT_{IMP\text{-}AM}}$ | 95.62% | **86.73%** | 75.61% | 74.85% | 75.91% | 76.12% | 74.24% | **74.13%** |

## C.3 Training details

The classifier is trained using SGD with momentum 0.9, weight decay $2 \times 10^{-4}$, and batch size 64. The initial learning rate is 0.1, which is reduced to 0.01 in the 75-th epoch. We stop training after 76 epochs. For $\mathrm{ADT_{EXP}}$, we adopt Adam [9] for optimizing the distribution parameters $\phi_i$. We set the learning rate for $\phi_i$ as 0.3, the momentum as $(0.0, 0.0)$, the number of optimization steps as $T = 7$, and the number of MC samples to estimate the gradient in each step as $k = 5$. For $\mathrm{ADT_{EXP\text{-}AM}}$ and $\mathrm{ADT_{IMP\text{-}AM}}$, we use only one MC sample for gradient estimation and use Adam with momentum $(0.5, 0.999)$ and learning rate $2 \times 10^{-4}$ to optimize the parameter $\phi$ of the generator network. We also adopt Adam with learning rate $2 \times 10^{-4}$ to optimize the parameter $\psi$ of the introduced variational in $\mathrm{ADT_{IMP\text{-}AM}}$.

## C.4 Baselines

Our primary baselines include: 1) standard training on the clean images (***Standard***); 2) adversarial training on the PGD adversarial examples (***AT_PGD***) [13]. Standard and $\mathrm{AT_{PGD}}$ are trained with the same configurations specified above. For training $\mathrm{AT_{PGD}}$, we perform PGD with $T = 7$ steps, and step size $\alpha = \epsilon/4$, which are the same as in [13]. On CIFAR-10, we incorporate several additional baselines, including: 1) the pretrained $\mathrm{AT_{PGD}}$ model (***AT_PGD***[†]) released by [13]; 2) adversarial training on the targeted FGSM adversarial examples (***AT_FGSM***) [11]; 3) adversarial logit pairing (***ALP***) [8]; and 4) feature scattering-based adversarial training (***FeaScatter***) [16]. We implement $\mathrm{AT_{FGSM}}$ and ALP by ourselves using the same training configuration specified above and use the pretrained model of FeaScatter. Note that all of these models have the same network architecture for a fair comparison.

## C.5 A feature attack for white-box evaluation

We incorporate a feature attack (FeaAttack) [12] for white-box robustness evaluation in this paper. The algorithm of FeaAttack is introduced below. Given a natural input $\mathbf{x}$, FeaAttack first finds a target image $\mathbf{x}'$ belonging to a different class. It minimizes the cosine similarity between the feature representations of the adversarial example and $\mathbf{x}'$ as

$$\boldsymbol{\delta}^* = \arg\min_{\boldsymbol{\delta} \in \mathcal{S}} \mathcal{L}_{cos}(f'_{\boldsymbol{\theta}}(\mathbf{x} + \boldsymbol{\delta}), f'_{\boldsymbol{\theta}}(\mathbf{x}')),$$

where $f'_{\boldsymbol{\theta}}(\cdot)$ returns the feature representation before the global average pooling layer for an input, and $\mathcal{L}_{cos}$ is the cosine similarity between two features. FeaAttack solves this objective function by

$$\boldsymbol{\delta}^{t+1} = \Pi_{\mathcal{S}}\big(\boldsymbol{\delta}^t - \alpha \cdot \mathrm{sign}(\nabla_{\mathbf{x}}\mathcal{L}_{cos}(f'_{\boldsymbol{\theta}}(\mathbf{x} + \boldsymbol{\delta}^t), f'_{\boldsymbol{\theta}}(\mathbf{x}')))\big).$$

$\boldsymbol{\delta}^0$ is initialized uniformly in $\mathcal{S}$. In our experiments, we set $\alpha = \epsilon/8$ and the number of optimization steps as 50. For each natural input, we randomly select 200 target images to conduct 200 attacks, and report a successful attack when one of them can cause misclassification of the model.

Table 10: Classification accuracy of TRADES and the three ADT-based methods trained with the TRADES loss on CIFAR-10 under white-box attacks with $\epsilon = 8/255$. We mark the best results for each attack and the overall results that outperform the baselines in **bold**, and the overall best result in **blue**.

| Model | $\beta$ | $\mathcal{A}_{\text{nat}}$ | FGSM | PGD-20 | PGD-100 | MIM | C&W | FeaAttack | $\mathcal{A}_{\text{rob}}$ |
|---|---|---|---|---|---|---|---|---|---|
| TRADES | 1.0 | 87.99% | 57.67% | 51.08% | 48.41% | 53.32% | 49.29% | 51.07% | 47.75% |
| ADT$_{\text{EXP}}$ | 1.0 | **89.74%** | 59.47% | 52.39% | 49.88% | 54.74% | 50.75% | 51.29% | **49.05%** |
| ADT$_{\text{EXP-AM}}$ | 1.0 | 88.86% | 62.89% | **54.44%** | **51.66%** | **56.09%** | **52.33%** | **54.61%** | **50.78%** |
| ADT$_{\text{IMP-AM}}$ | 1.0 | 88.80% | **68.35%** | 54.22% | 51.09% | 54.95% | 51.84% | 54.19% | **50.14%** |
| TRADES | 6.0 | 84.02% | 60.08% | 56.06% | 54.49% | 57.27% | 53.62% | 55.18% | 52.64% |
| ADT$_{\text{EXP}}$ | 6.0 | 84.66% | 61.72% | 57.71% | **56.17%** | **58.74%** | **55.16%** | 56.65% | **54.21%** |
| ADT$_{\text{EXP-AM}}$ | 6.0 | 84.85% | 66.09% | 57.67% | 55.73% | 58.38% | 54.79% | 58.94% | **54.09%** |
| ADT$_{\text{IMP-AM}}$ | 6.0 | **84.96%** | **68.34%** | **57.82%** | 55.45% | 58.58% | 54.36% | **59.01%** | 53.66% |

# D  Supplementary experimental results

We provide more experimental results in this section.

## D.1  Full results on CIFAR-100 and SVHN

We provide the full experimental results of Standard, AT$_{\text{PGD}}$, ADT$_{\text{EXP}}$, ADT$_{\text{EXP-AM}}$, and ADT$_{\text{IMP-AM}}$ under all adopted white-box attacks on CIFAR-100 and SVHN in Table 9.

## D.2  Full results on TRADES

In TRADES [17], the minimax optimization problem is formulated as

$$\min_{\boldsymbol{\theta}} \frac{1}{n} \sum_{i=1}^{n} \Big\{ \mathcal{L}(f_{\boldsymbol{\theta}}(\mathbf{x}_i), y_i) + \beta \cdot \max_{\boldsymbol{\delta}_i \in \mathcal{S}} \mathcal{D}_{\text{KL}}(f_{\boldsymbol{\theta}}(\mathbf{x}_i + \boldsymbol{\delta}_i), f_{\boldsymbol{\theta}}(\mathbf{x}_i)) \Big\},$$

where $\beta$ is a hyperparameter balancing the trade-off between natural and robust accuracy. The full experimental results of TRADES and the three variants of ADT when integrated with the TRADES loss are shown in Table 10. We evaluate their performance by all adopted white-box attacks and report the worst-case robustness as in Eq. (12).

## D.3  Convergence of learning the explicit adversarial distributions

We study the convergence of the explicit adversarial distributions introduced in Sec. 3.1 by attacking AT$_{\text{PGD}}$ and ADT$_{\text{EXP}}$ with varying iterations. We set the learning rate of $\phi_i$ as 0.3, the momentum as $(0.0, 0.0)$, the number of MC samples to estimate the gradient in each step as $k = 10$, and vary the attack iterations from 0 to 100. We show the classification loss and accuracy in Fig. 6. Learning the explicit adversarial distributions can converge soon within a few iterations.

Figure 6: Classification loss (i.e., cross-entropy loss) and accuracy (%) of AT$_{\text{PGD}}$ and ADT$_{\text{EXP}}$ under the explicit adversarial distributions attack with different attack iterations.

## D.4 Training time

We provide the one-epoch training time of Standard, $AT_{PGD}$, $ADT_{EXP}$, $ADT_{EXP-AM}$, and $ADT_{IMP-AM}$ on CIFAR-10 in Fig. 7. As can be seen, $ADT_{EXP}$ is nearly $5\times$ slower than $AT_{PGD}$ since we use $k = 5$ MC samples to estimate the gradient w.r.t. the distribution parameters in each step. Nevertheless, by amortizing the adversarial distributions, $ADT_{EXP-AM}$ and $ADT_{IMP-AM}$ are much faster than $ADT_{EXP}$, and nearly $2\times$ faster than $AT_{PGD}$.

Figure 7: The training time (s) for one epoch of Standard, $AT_{PGD}$, $ADT_{EXP}$, $ADT_{EXP-AM}$, and $ADT_{IMP-AM}$ on CIFAR-10.

## D.5 Comparison with Chen et al. [1]

We further compare ADT with the L2L framework in [1]. Their method is similar to ours in the sense that they also adopt a generator network to produce adversarial examples, and perform adversarial training on those generated adversarial examples. The essential different between our methods and theirs is that we propose an adversarial distributional training framework to learn the distributions of adversarial perturbations, while their method is a variant of the vanilla adversarial training with a different approach to solving the inner maximization.

Since the source code is not provided by Chen et al. [1], we tried to reproduce their reported results with the same training configuration specified in their paper, but we failed. Therefore, we adopt the same configuration as in ADT for training the L2L model. Table 11 shows the results of L2L, $ADT_{EXP-AM}$, and $ADT_{IMP-AM}$, which use the same classifier architecture and generator network. Our ADT-based methods outperform L2L in most cases, showing the advantages of learning the distributions of adversarial perturbations upon finding a single adversarial example.

Table 11: Classification Accuracy of L2L [1], $ADT_{EXP-AM}$, and $ADT_{IMP-AM}$ on CIFAR-10 under white-box attacks with $\epsilon = 8/255$.

| Model | L2L | $ADT_{EXP-AM}$ | $ADT_{IMP-AM}$ |
|---|---|---|---|
| $\mathcal{A}_{nat}$ | **88.15%** | 87.82% | 88.00% |
| FGSM | **65.50%** | 62.42% | 64.89% |
| PGD-20 | 48.55% | 51.95% | **52.28%** |
| PGD-100 | 47.14% | **51.26%** | 51.23% |
| MIM | 49.03% | **52.99%** | 52.64% |
| C&W | 49.22% | 51.75% | **52.65%** |