[Reviews · NeurIPS 2020]

Review 1

Summary and Contributions: The paper proposes a modification to adversarial training where in the inner maximization, the adversary picks a paramerized distribution of examples. The goal is to increase the diversity of the attack in order to boost the performance of the resulting defense.

Strengths: The paper proposes a novel idea and the idea is presented well. The diagrams were also helpful in understanding the idea and the writing was clear. The use of the explicit generative model + reparameterization trick was also very cool to see in the context of adversarial examples. I also appreciate the authors' thoroughness in understanding the method: comparisons with TRADES were done, as well as ablation studies around different components of the algorithm. The authors also provided thorough documentation of how adversarial evaluation was done, which is rare and highly appreciated.

Weaknesses: The biggest flaw with the paper, in my opinion, is the empirical results section. Specifically, the results for AT_PGD seem below the state of the art reported by many other works (e.g. the pre-trained models here: https://github.com/MadryLab/robustness, or the models here: https://arxiv.org/abs/2002.11569---the latter reports 56% robust accuracy and 87% natural accuracy for a smaller architecture than the ones tried here). The lack of confidence intervals/multiple trials also makes it difficult to ascertain the significance of the results.

Correctness: The results are correct to the best of my knowledge.

Clarity: The paper is very well written (there are a few spelling errors/mixed up words but not enough to hinder reading too much---I would recommend a round of copy editing.)

Relation to Prior Work: Yes (though I may not be aware of all the relevant works to cite).

Reproducibility: Yes

Additional Feedback: I thought the method was very cool! One thing that I thought the paper was doing (which turned out to be a misunderstanding, I think) is relaxing the l2 adversarial constraint bit. This is more of an intuition (and did not affect my review in any way), but to some extent is seems like if what one cares about is L2-adversarial robustness, then maximizing the inner loss with PGD is in some sense going to be "optimal"/hard-to-beat (some results in the Madry et al paper corroborate this, few-step PGD is pretty good at finding the best maxima we can find in general.) On the other hand, what you have is a weaker adversary (the distributional one + entropic regularizer), but it has the advantage of being a potentially structured way of enforcing a better constraint than L2 robustness. Again this isn't part of my review, but it would be cool to see if it is possible to define a new robustness constraint that is explicitly tailored to your learned adversary (e.g. some kind of statistical similarity) that better leverages your attack approach. ---- Post-rebuttal comment: I have read the authors rebuttal and it has successfully addressed many of the points in my review. I would have liked for the error bars to be 95% confidence intervals rather than +/- stdev, but this is a more minor concern and I have raised my score by one. Thank you for a well-written and thorough rebuttal!


Review 2

Summary and Contributions: ************************************************************************ Update after author response: I have read the author's response to my questions. I am still not sure how much to believe that the proposed method will be more robust towards new, unseen methods in the future. From a theoretical point of view I am not entirely sure why entropic regularization should select a good trade-off between diversity and effectiveness of the training attacks and from a practical point of view the "cat-and-mouse" phenomenon makes it difficult to jugde. However, these concerns are inherent to adversarial defences as a field and the author's have done all that can be expected to address them, which is why I believe that the paper should be accepted. ************************************************************************ In this work, the authors propose to extend adversarial training by learning,distributions of adversarial examples as opposed to a single such example. By using an entropy regularization, they enforce diversity among these examples and argue that it leads to improved robustness to different attacks. A thorough suite of experiments substantiates these claims.

Strengths: The idea of randomizing adversarial examples is natural and well motivated. The experiments, as well as the theoretical treatment of the gradient computation seem to be done thoroughly.

Weaknesses: The authors argue for distributional adversarial training by pointing out that competing method are only trained with respect to one attack that is used . This seems to be a bit misleading/exagerated, since adversarial distributional training is similarly only trained by one attack, only that this attack happens to operate on probability measures instead of individual samples. The key question is to what extend the entropic regularization allows to find classes of examples that, while adversarial, are sufficiently diverse to make allow for robustness to adversarial examples obtained by other methods of attack. The numerical evidence support this claim somewhat in that the proposed method (Adversarial Distributional Training, ADT) provides some improvement over some of the baselines while those baselines that decisively beat ADT have at least one attack that they perform poorly on. However, it is not ruled out that another attack might be developed that performs well against ADT, only time will tell the effectiveness of ADT to this respect.

Correctness: As far as I can tell (no expert in adversarial robustness), the methodology seems sound

Clarity: The paper is well written

Relation to Prior Work: As far as I can tell (no expert in adversarial robustness), related work is discussed appropriately.

Reproducibility: Yes

Additional Feedback: When comparing to the performance of past defences, such as FeaScatter it would be helpful to know if the attacks that it struggles on were known at the time FeaScatter was conceived. If this was not the case, it weakens the argument towards ADT being more robust since, relative to the known attacks at the time of writing, it performs worse than FeaScatter. Being clear about the attacks known when each of the baseline methods were proposed would help clarify to what extent improvements might be caused by a benefit of hindsight


Review 3

Summary and Contributions: This paper proposed a uniform method to learn an adversarially robust model. Instead of using some specified adversarial samples, this paper concluded in an entropy regularization term.

Strengths: 1. This paper provided an interesting insight into adversarially robust learning. Addressing adversarial learning by worst-case optimization is novel. 2. Tenicholy sounds. 3. Experiments conducted on several datasets have been well discussed. 4. Both explicit and implicit methods are proposed. 5. The code is available for better reproducibility.

Weaknesses: Some similar works that also discuss worst-case distribution should be discussed in related work.

Correctness: Yes.

Clarity: This paper is well written and easy to read.

Relation to Prior Work: Yes.

Reproducibility: Yes

Additional Feedback:


Review 4

Summary and Contributions: This paper brings variational distribution and the concept of reparameterization trick to the task of adversarial training (AT): adversarial distributional training (ADT). The variational distribution enhanced the robustness of AT up to their distributional coverage, yielding more robust results compared to non-distributional versions of AT. # updates after author response My questions are well addressed in the authors' response, I appreciate the authors for the clarification.

Strengths: While existing works focus on the inner optimization process and solve the minimax problem of AT in an alternate fashion. The proposed method enables solving the minimax problem simultaneously. ADT allows the model to see various adversarial noise-injected examples while training and eventually makes the model robust to all adversarial attacks if they are covered in variational distribution.

Weaknesses: At line 88, the authors mentioned the inner maximization problem can be easily solved with a degenerated solution of ADT: Dirac one. However, in figure 5, the model with no entropy regularization (lambda = 0) shows comparable results with those used the regularization. Maybe because the amortized version needs not to be degenerated to Dirac one to solve the inner maximization to match the term for data points? I think the results from the non-amortized version should be compared to understand the effect of the regularization term.

Correctness: I found the claims and methods are correct. But as I am not used to this area of adversarial training, I humbly admit that my judgment could be wrong.

Clarity: I found the paper is easy to follow.

Relation to Prior Work: This paper delivers detailed explanations of previous methods and clearly stated their direction of research: the use of variational distribution for AT, which is not found in previous approaches.

Reproducibility: Yes

Additional Feedback:

[Author Response · NeurIPS 2020]

We thank all reviewers for their helpful and constructive comments. We'll further improve the paper in the final version. Below we address their detailed comments.

**R1: The results for $AT_{PGD}$ seem below the state-of-the-art:** We need to clarify that the $AT_{PGD}$ model is trained by following the experimental settings in [36]. We found that the training configuration of the state-of-the-art $AT_{PGD}$ in [*1] pointed out by R1 differs from [36] in several aspects,

Table A: Model accuracy (%) on CIFAR-10 following [*1].

| Model | $\mathcal{A}_{\mathrm{nat}}$ | PGD-10 | PGD-20 | PGD-100 |
|---|---|---|---|---|
| $AT_{PGD}$ | 86.41 | 55.90 | 54.52 | 54.20 |
| $ADT_{EXP}$ | 86.49 | **56.84** | **55.43** | **55.01** |
| $ADT_{EXP-AM}$ | 87.27 | 56.28 | 54.88 | 54.58 |
| $ADT_{IMP-AM}$ | **87.38** | 56.63 | 55.10 | 54.43 |

including early stopping, weight decay factor, and the number of PGD steps. We also need to point out that the model which achieves $56\%$ robust accuracy and $87\%$ natural accuracy in [*1] is a Wide-ResNet-34-10 model (Table 1 in [*1]). Their smaller model (i.e., PreActResNet18) achieves $53\%$ robust accuracy (Table 2 in [*1]). Besides, the robust accuracy is evaluated by PGD-10 in [*1], which is a weaker adversary than we used in experiments. To fairly compare with the state-of-the-art, we reproduce the results of [*1] and train ADT based models using the same settings/hyperparameters as in [*1]. The results of those models on CIFAR-10 are shown in Table A. By using the same training settings, our models can also improve the performance over $AT_{PGD}$. We'll include the results in the final version.

**R1: Confidence intervals/multiple trials:** In Table B, we show the mean and standard deviation of accuracy of $AT_{PGD}$ and ADT based models over 3 runs (using the submitted code). The standard deviation is small compared with the performance gap. We'll include the full results in the final version.

Table B: Model accuracy (%) on CIFAR-10 over 3 runs.

| Model | $\mathcal{A}_{\mathrm{nat}}$ | PGD-20 | PGD-100 |
|---|---|---|---|
| $AT_{PGD}$ | 86.50±0.14 | 49.77±0.21 | 49.34±0.27 |
| $ADT_{EXP}$ | 87.15±0.13 | 52.38±0.23 | 51.89±0.22 |
| $ADT_{EXP-AM}$ | 87.30±0.09 | 53.01±0.22 | 52.45±0.28 |
| $ADT_{IMP-AM}$ | 87.58±0.14 | 51.90±0.15 | 50.94±0.16 |

**R1: $\ell_2$ adversarial constraint:** We need to clarify that we consider the $\ell_\infty$ norm constraint in this paper. However, our methods can be easily extended to the $\ell_2$ norm. We agree that PGD is effective to find local maxima of the inner problem, but we show in Fig. 1 that the adversarial distributions can better explore the space of possible perturbations and characterize more diverse adversarial examples, resulting in more robust models, as discussed in Sec. 2.2.1.

**R1: A new robustness constraint:** Thanks for the insightful comment. We think that the proposed ADT framework is flexible to integrate a new robustness constraint. We'll consider this in future work.

**R2: ADT is trained by one attack that operates on probability measures instead of individual samples:** Yes, ADT uses a single attack which can find a distribution of adversarial examples instead of an individual sample. We have discussed in Sec. 2.2.1 the superiority of our approach upon others which generate individual adversarial examples by a single attack. We'll further polish our arguments in the final version to make them not misleading.

**R2: To what extend the entropic regularization allows to find adversarial and sufficiently diverse examples:** When using no entropic regularization, ADT degenerates into AT such that the adversarial examples are not diverse. When using a very large entropic regularization, the generated examples are diverse, but are not adversarial enough. Thus, we use a hyperparameter $\lambda$ to control the strength of the entropy term in Eq. (5). As it's hard to derive the optimal value for $\lambda$, we did an ablation study on the effects of $\lambda$ in Fig. 5. Our results suggest that choosing an appropriate $\lambda$ (e.g., 0.01) can ensure the generated examples being both adversarial and diverse for learning a robust model.

**R2: Another attack might be developed that performs well against ADT:** Just like other empirical defenses, we cannot guarantee that there aren't any attacks that can beat our defenses. However, we have tried our best to evaluate the robustness of our defenses, including adopting a plenty of attacks, calculating the per-example accuracy, evaluating black-box attacks, and visualizing the loss landscape. Experiments suggest that the common failure modes [2,6,56] of previous defenses do not occur in our method. We'll also release our code and pre-trained models for future evaluations.

**R2: Being clear about the attacks known when each of the baseline methods were proposed:** One of the challenges of adversarial robustness research is that there exists a "cat-and-mouse" game between attacks and defenses, i.e., the defenses were later shown to be ineffective against new attacks, which has drawn much attention in this field [2,6,56]. Therefore, it's important to develop robust models that not only are robust to existing attacks but can also generalize to new ones [49], which is also the main motivation of our work. Although FeaAttack was proposed later than FeaScatter, it can also prove the ineffectiveness of FeaScatter. As above, we have tried our best to evaluate the worst-case robustness of our defenses following the guidelines in [6], and we're willing to test our models by future attacks continuously. We do believe that our defenses can generalize to new attacks better than the baselines.

**R3: Related works on worst-case distribution:** Thanks for the suggestion. We'll discuss them in the final version.

**R4: The degenerated solution of ADT:** When $\lambda = 0$, the adversarial distribution degenerates into a Dirac distribution and ADT becomes AT. So we expect that the performance of ADT ($\lambda = 0$) matches the performance of $AT_{PGD}$. As can be seen from Fig. 5, the model trained with $\lambda = 0$ gets about $50\%$ accuracy against attacks, which is similar to the results of $AT_{PGD}$. But with the entropic regularization, ADT obtains more than $2\%$ accuracy improvements, as shown in Fig. 5. We'll also show the results of $ADT_{EXP}$ with different $\lambda$ in the final version.

[*1] L. Rice, E. Wong, J.Z. Kolter. Overfitting in adversarially robust deep learning. ICML 2020.


[Meta-Review · NeurIPS 2020]

Thank you for your submission to NeurIPS. After discussion, the reviewers are all in agreement that the proposed method does present an interesting and significant addition to the literature on adversarial training. The one criticism that the reviewers raised, that the method did not compare to the current state of the art in standard adversarial training, was well-addressed by the author response, and I'd strongly encourage them to include these results in the final version.